# Home Indoor Environmental Quality and Attention Deficit Hyperactivity Disorder

**Sima Alizadeh** [1,*] , **Catherine E. Bridge** [1] , **Bruce H. Judd** [1] **and Valsamma Eapen** [2]

1 School of the Built Environment, Faculty of Arts, Design and Architecture, University of New South Wales, Sydney, NSW 2052, Australia

2 School of Psychiatry-SWSAHS, Faculty of Medicine and Health, University of New South Wales, Sydney, NSW 2052, Australia

* Correspondence: s.alizadeh@unsw.edu.au

**Abstract:** Indoor environmental quality (IEQ) has been found to influence children's health and behaviour, particularly conditions, such as attention deficit hyperactivity disorder (ADHD). This research aimed to ascertain whether housing IEQ impacted the symptoms of ADHD and its related behaviours. This study involved 435 parents of children and adolescents (aged 5–17) with ADHD residing in Australia. We utilised an online survey tool comprising the home version of the ADHD Rating Scale for Children and Adolescents and the Housing Environmental Quality Assessment Tool. The findings revealed that all the housing IEQ factors were associated with both the diagnosis and the severity of the symptoms. The results indicated that, for over one in ten children with ADHD (13.3%), the diagnosis was impacted by a collective contribution of air quality, acoustic quality, lighting quality, and the thermal comfort within their home. The air quality and the thermal comfort predicted a 4% variance in the severity of inattention. Additionally, air and lighting qualities predicted a 9.7% variance in the severity of hyperactivity-impulsivity, and all the factors of air quality, acoustic quality, lighting quality, and thermal comfort predicted a 10.9% variance in the severity of combined ADHD. In conclusion, this research provided insights into the importance of housing features and suggests that improving the housing indoor environmental quality, mainly thermal comfort, air, and lighting quality, could positively correlate with alleviating ADHD symptoms and severity among children and adolescents. Therefore, ensuring an appropriate indoor environmental quality should be prioritised in housing design, modification, and building, especially for those with ADHD.

**Keywords:** ADHD; housing; environmental quality; children and adolescents; Australia

## 1. Introduction

The indoor Environmental Quality (IEQ) is the environmental conditions inside a building and their impacts upon its occupants' mental and physical health, comfort, satisfaction, productivity, safety, and security [1,2]. These conditions are also considered significant factors influencing the occupants' assessment of their quality of life [3] and include air quality, noise, lighting, and thermal comfort [4]. Therefore, the IEQ has become a significant public health issue.

The IEQ has attracted more attention as people have been staying indoors longer [5]; especially during the recent COVID-19 pandemic. Due to lockdowns, quarantine, isolation, and other social restrictions, people remained indoors even longer than in pre-covid times, thereby having more exposure to the influences of their indoor environmental quality [6,7]. Research has shown that generally, Australians spend around 90% of their time indoors [8]. Children, while growing up, also spend most of their time (approximately 70–85%) within their home environment [9–11].

Wyon and Wargocki [12] reported that the adverse impacts of a poor indoor environment could be up to 10% for adults and over 20% for children. This shows that children are

more sensitive to the adverse influences of their indoor environmental quality while their organs and tissues are still actively developing [13,14].

However, most existing Australian housing was designed and built considering the average user: a healthy and young adult man [15–17]. Building regulations, therefore, need to be better developed in recognising people with different abilities. This is especially so for children with different cognitive abilities, such as those diagnosed with attention-deficit/hyperactivity disorder (ADHD) who are extra sensitive to the impacts of their everyday surroundings and home environment [18].

To address this gap, this project aimed to identify any housing indoor environmental qualities adversely impacting children's ADHD symptoms and behaviours. The findings are expected to contribute to the principles of potential housing characteristics, providing as supportive an environment as possible for these children.

ADHD is the most frequently diagnosed behavioural and neurodevelopmental disorder in childhood [19], characterised by one or more core symptoms of inattention, impulsivity, and hyperactivity [20,21]. As many as 75% of children with ADHD are likely to report at least one other psychiatric condition (also referred to as a comorbid condition) [22,23], such as anxiety, depression, sleep disorders, learning disabilities, and autism [24–27].

Today, ADHD impacts more children worldwide and occurs across different cultures, races, socioeconomic status, and intelligence levels [28]. The prevalence of ADHD has risen substantially in recent decades, labelled by the Centers for Disease Control and Prevention (CDC) as "a serious public health problem" [29]. A meta-regression analysis of international epidemiological research has shown a worldwide pooled ADHD prevalence of 5.3% in 2007 [30]. While a more recent meta-analysis of 175 research studies worldwide in 2015 found a growth in ADHD prevalence to 7.2% (roughly 129 million) among children aged 18 and under [31].

Children with ADHD are extra sensitive to their environment [32]. Because of their sensory dysfunction, they are also influenced by their physical and social environments, such as through emotional, visual, and auditory effects [33–35]. That is, they experience significant difficulty remaining focused in an environment with various sensory inputs, which has a far-reaching impact on their capabilities and mental wellbeing [18]. This suggests that the home's indoor environmental quality can significantly impact their health and daily life.

Exposure to a poor IEQ can cause short-term (e.g., allergies, asthma) and long-term (e.g., cardiovascular problems, acute respiratory disease, and skin infection) effects [1,36–39], with links to physical and mental health impacts [40]. Behaviour toxicity is also caused by the undesirable effects of a poor IEQ, such as indoor air pollutants and certain building materials that release chemicals or the use of gas for cooking [41,42].

Several behavioural toxins and hazardous materials (e.g., lead, mercury, and manganese) impact psychological wellbeing and are linked to the symptoms of anxiety and depression [43]. Many studies about behavioural toxins, including heavy metals, pesticides, and solvents, have also centred around neurological and cognitive effects [44]. An example is cognitive deficits related to early childhood lead exposure [45]. Lead obstructs self-regulatory behaviour in children (such as focused concentration and frustration tolerance), which is associated with behavioural issues and conduct disorder (e.g., yelling, fighting, and other forms of aggression) [46,47]. Exposure to heavy metals (e.g., lead, manganese, zinc, and molybdenum) have also been explicitly linked to ADHD and learning disorders in school-aged children [48–50].

A meta-analysis of 27 articles with over three million subjects about air pollutants caused by smoke found that even prenatal exposure to maternal smoking during pregnancy or smoking termination during the first three months of pregnancy were significantly associated with childhood ADHD, after controlling for socioeconomic status and parental psychiatric history [51]. High levels of carbon dioxide in the air, produced by human respiration, can cause headaches, drowsiness, and concentration problems [52]. A study on eight different European schools (n $\approx$ 800) observed that higher carbon dioxide levels in

classrooms, caused by poor ventilation, reduced students' concentration and performance on tests and increased students' complaints of health issues [53].

Noise exposure also disrupts health and mental wellbeing. Research has found a dose-response relationship between traffic noise and psychological distress among children [54]. It has also revealed that airport noise adversely influenced children's and adults' psychological wellbeing [55,56]. Other studies have also shown that different sound levels can affect children's ability to concentrate [57–59], and lower noise levels can increase children's capacity to focus [60]. More specifically, research has indicated the significance of acoustic quality for people with neurological, attention, and cognitive impairments, such as ADHD and autism spectrum disorder (ASD). For example, Tufvesson [61] found that one of the environmental factors affecting children's attention was interior acoustic issues, such as background noise and sound filtration (e.g., sound filtrated into a room). Mostafa [62] mentioned that if a space requires high acoustical quality, such as a bedroom or study room, it may be preferable to reduce the openings in number and size. The windows themselves should be double- or triple-glazed.

In this regard, Evans [63] showed that levels of illumination, especially the amount of daylight exposure, affects psychological wellness. In a study, the design considerations of an ideal classroom environment for children with ASD, high-level glazing windows (to prevent unwanted sunlight and glare), lighting type (dimmable, not fluorescent), volumetric expression, background noise, and quiet rooms were found to be highly significant [64,65]. Studies have revealed that both artificial light [34,66,67] and daylight [61,68] impact the attention and behaviour of children. Lighting is also referred to in several 'autism-friendly' design guidelines, most frequently about avoiding fluorescent lighting and glare [69–72]. Similarly, Mostafa [62] showed that the type of artificial lighting is important, and those with ADHD report a high sensitivity to strong light. Other research has also found that classrooms with exposed fluorescent lighting increased hyperactive behaviours (fidgeting and lack of concentration) among children with ADHD [73]. Both the flickering light and the low humming sound that fluorescent lighting emits caused visual and auditory issues creating a disturbing luminous environment for these children and increasing their hyperactive behaviours [62].

Research has shown that natural light, especially morning light, is preferable to provide adequate lighting in an area for children with ADHD to recover from their symptoms. Morning light therapy improves circadian rhythm sleep disorders and shifts sleep toward an earlier circadian timing [74]. Cross-states (49 U.S. states) and multinational (nine non-U.S. countries) studies found that the prevalence of ADHD is less in zones with higher exposure to solar intensity (particularly morning light) after controlling for demographic, socioeconomic, and perinatal (e.g., low birth weight and infant mortality) factors [75]. However, lighting should be used indirectly to reduce distractibility [62,76]. Direct or too much daylight has been shown to cause distraction and interrupt children's concentration, especially in areas where focus and attention are essential [77].

A few studies have examined the links between mental health and thermal comfort. In a study on European housing, Bonnefoy et al. [78], found a relationship between depression/anxiety and residing in a home with insufficient protection against exposure to external factors, such as cold and draughts, in addition to noise and lack of light. Research on 21 Dutch school buildings revealed that children's health and comfort in their classroom environment were mainly influenced by noise and smell, followed by sunlight and temperature [79]. Higher temperatures and humidity increased students' reports of discomfort, and their attainment and task performance declined as concentration spans reduced [80]. Regarding children with ADHD, it has been assumed that a hyperactive individual may require a cooler environment than a less active or sedentary one [62]. However, not enough research has been performed on ADHD and thermal comfort.

## 2. Materials and Methods

This study employed an observational design where the ADHD diagnosis outcome and the IEQ exposures of the survey participants were examined to assess the indoor environmental quality within their home setting, targeting the population of children and adolescents with ADHD in Australia. The data were collected from 435 participants through a random sampling technique by administering an anonymous (self-administered) online questionnaire on relevant organisations' web pages, social media, and online community platforms. Participants were parents (including a few legal guardians and carers) of children and adolescents (5–17 years of age) with a diagnosis of ADHD residing in Australia.

Two instruments were used in this study for data collection: (1) the Housing Environmental Quality Assessment tool (HEQAT) for measuring the home indoor environmental quality, including air quality, acoustic quality, lighting quality, and thermal comfort; and (2) the ADHD Rating Scale-IV for Children and Adolescents (Home Version) [81] to evaluate the diagnosis of ADHD and the severity of the symptoms among children and adolescents. The core symptoms of ADHD diagnosis were evaluated, including inattention (IA), hyperactivity/impulsivity (HI), and combined ADHD type (COM), as well as ADHD with comorbid conditions that the parents reported.

A control group was also included in this study for comparison. It comprised children and adolescents whose ADHD scores on the ADHD Rating Scale did not reach the cut-off point of inattention and/or hyperactivity-impulsivity scores based on the user manual guide for scoring. The severity of ADHD symptoms was measured in percentiles.

The research analyses were carried out using IBM SPSS Statistics software. The Chi-Square ($x^2$) test or Fisher's exact test and the one-way ANOVA test were used to compare the difference in the housing environmental quality factors associated with the diagnosis of ADHD symptoms. The Univariate analyses of variances and Pearson's Correlation Coefficient were conducted to explore the relationship between the housing environmental quality factors and the severity of the ADHD symptoms. The Multinomial Logistic Regression and Stepwise Linear Regression analyses were performed to evaluate the predictive power of the IEQ factors for ADHD diagnosis and ADHD severity and to explore the most significant contributing co-variates within the regression models.

## 3. Data Analyses and Results

The housing IEQ was measured by both objective and subjective format questions relating to four aspects: air quality, acoustic quality, lighting quality, and thermal comfort. To explore their relationship with an ADHD diagnosis and the severity of the symptoms (as dependent variables), 13 items were entered into the analyses. These items studied each aspect of the IEQ from different viewpoints, including the ratings of their suitability and the level of controllability. The associations of the relevant issues were also explored. Subjective-format questions, such as the rate of suitability were measured using five-point Likert scales with the choices from very unsuitable to very suitable. Objective-format questions, such as the level of controllability or relevant issues were measured using multiple-choice questions and required participants to identify attributes or conditions at home, such as the presence or absence of specific items. Additionally, the direction of the windows of the home's main living area were considered in evaluating the quality of natural lighting and the amount of sunlight exposure in the home. However, the studied housing IEQ parameters were not measured on-site by the researchers as the survey was conducted in the form of an anonymous (self-administered) online questionnaire.

Before conducting an inferential analysis, a descriptive analysis was performed, and the normality of the data distribution was tested. Table 1 presents the descriptive analysis, including the frequency statistics of the research variables indicating a symmetrical distribution of the data.

**Table 1.** Descriptive analysis: frequency statistics of research variables.

| Research Variables | Central Tendency | | | Dispersion | | | Distribution | |
|---|---|---|---|---|---|---|---|---|
| | Mean | Median | Mode | Std. Deviation | Variance | Range | Skewness | Kurtosis |
| ADHD Diagnosis | 2.63 | 3.00 | 3.00 | 1.17 | 1.37 | 4 | −0.925 | −0.054 |
| IA Severity | 3.29 | 3.31 | 3.66 | 0.951 | 0.905 | 4.52 | 0.148 | −0.038 |
| HI Severity | 3.34 | 3.47 | 3.66 | 1.24 | 1.55 | 5.83 | −0.475 | 0.089 |
| COM Severity | 3.59 | 3.47 | 5.29 | 1.04 | 1.09 | 4.74 | 0.077 | −0.677 |
| IEQ | 3.66 | 3.60 | 3.60 | 0.643 | 0.414 | 3.20 | −0.059 | −0.239 |

### 3.1. ADHD and Air Quality

There was a borderline significant relationship between ADHD diagnosis in children and the air quality issues within the home (Fisher's Exact Test = 8.56, $p = 0.056$), indicating that children with ADHD and comorbid conditions were more likely to report unpleasant odours (14.1%) and less likely to report the absence of unpleasant odours (85.9%) (Figure A1). It was also found that children of parents who reported stuffy or stale air within the home environment scored lower on the severity of inattention when compared to children of parents who did not report such air conditions (Table 2, Figure A2).

**Table 2.** Relationship between air quality and severity of ADHD symptoms.

| Air Quality | ADHD Severity | N | df | F | Mean Difference | Std. Error | Sig. |
|---|---|---|---|---|---|---|---|
| Stuffy or stale air | Inattention | 387 | 1 | 4.91 | 0.275 | 0.124 | 0.027 |
| Vacuum cleaner | Inattention | 386 | 1 | 6.92 | 0.638 | 0.242 | 0.009 |
| | Hyperactivity-impulsivity | 385 | 1 | 18.56 | 1.379 | 0.320 | 0.000 |
| | Combined ADHD | 386 | 1 | 20.97 | 1.202 | 0.262 | 0.000 |
| Indoor plants | Hyperactivity-impulsivity | 385 | 1 | 3.74 | 0.251 | 0.130 | 0.054 |

Pairwise comparisons by Bonferroni.

Regarding the controllability of the air quality, there was a significant difference between using a vacuum cleaner in the home and the severity of inattention, hyperactivity-impulsivity, and combined ADHD type (Table 2). The severity of the ADHD symptoms was significantly higher in those residing in homes where a vacuum cleaner was not used when compared to homes where a vacuum cleaner was used (Figures A3–A5). Having indoor plants also showed a borderline significant association with a lower severity of hyperactivity-impulsivity, indicating that hyperactive-impulsive children who had indoor plants within their home showed milder symptoms than those who did not have indoor plants (Table 2, Figure A6).

### 3.2. ADHD and Acoustic Quality

The reported suitability of the acoustic quality was significantly lower within the homes of children with ADHD and comorbid conditions than in the control group (Table 3, Figure A7). Additionally, the severity of hyperactivity-impulsivity and combined ADHD type showed a very weak, negative, and significant relationship with parents' ratings of the suitability of the acoustic quality within their home, indicating that the severity of

these ADHD symptoms was significantly lower in children residing in homes with a more suitable acoustic quality (Table 3, Figures A8 and A9).

**Table 3.** Relationship between acoustic quality and ADHD diagnosis and severity of symptoms.

| Acoustic Quality | ADHD Diagnosis and Severity of Symptoms | N | df | F | Pearson Correlation (2-Tailed) | Mean | Std. Deviation | Std. Error | Sig. |
|---|---|---|---|---|---|---|---|---|---|
| Suitability of acoustic quality | Control group (no ADHD) | 378 | 4 | 3.13 | - | 3.67 | 0.994 | 0.182 | 0.015 |
| | ADHD with comorbid conditions | | | | | 3.18 | 0.951 | 0.069 | |
| | Severity of hyperactivity-impulsivity | 382 | - | - | −0.118 | - | - | - | 0.021 |
| | Severity of combined ADHD | 383 | - | - | −0.115 | - | - | - | 0.024 |

Post hoc test: Bonferroni.

### 3.3. ADHD and Lighting Quality

A borderline significant association was found between artificial lights flickering within the home and an ADHD diagnosis (Fisher's Exact Test = 8.39, $p = 0.052$). Children with ADHD and comorbid conditions were more likely to report artificial light flickering within the home environment (11.9%) and were least likely to report no artificial light flickering (88.1%) (Figure A10). Moreover, the reported suitability of the artificial lighting quality (or electrical lighting) showed a very weak, negative, and significant relationship with the severity of hyperactivity-impulsivity in children, indicating that the severity of hyperactivity-impulsivity was higher in children with an unsuitable artificial lighting quality within their home (Table 4, Figure A11).

**Table 4.** Relationship between lighting quality and severity of ADHD symptoms.

| Lighting Quality | ADHD Severity | N | df | F | Pearson Correlation (2-Tailed) | Mean Difference | Std. Error | Sig. |
|---|---|---|---|---|---|---|---|---|
| Suitability of artificial lighting quality | Hyperactivity-impulsivity | 368 | - | - | −0.118 | - | - | 0.024 |
| Reflection or glare | Hyperactivity-impulsivity | 378 | 1 | 4.88 | - | −0.420 | 0.190 | 0.028 |
| Too much daylight | Hyperactivity-impulsivity | 378 | 1 | 7.14 | - | −0.555 | 0.208 | 0.008 |
| | Combined ADHD | 379 | 1 | 6.03 | - | −0.435 | 0.177 | 0.015 |
| Insufficient artificial lighting | Combined ADHD | 379 | 1 | 3.52 | - | −0.315 | 0.168 | 0.062 |
| Windows' placement facing south vs. east * | Hyperactivity-impulsivity | 377 | 5 | 2.41 | - | 0.588 | 0.203 | 0.004 |
| | Combined ADHD | 377 | 5 | 2.35 | - | 0.478 | 0.172 | 0.006 |
| Windows' placement facing south vs. north * | Hyperactivity-impulsivity | 377 | 5 | 2.41 | - | 0.439 | 0.179 | 0.014 |
| | Combined ADHD | 377 | 5 | 2.35 | - | 0.318 | 0.151 | 0.036 |
| Windows' placement facing east vs. west * | Hyperactivity-impulsivity | 377 | 5 | 2.41 | - | −0.408 | 0.203 | 0.045 |
| | Combined ADHD | 377 | 5 | 2.35 | - | −0.441 | 0.171 | 0.010 |

Pairwise comparisons by Bonferroni. * Pairwise comparisons by LSD Test.

The analyses also revealed that the severity of hyperactivity-impulsivity was significantly higher in children whose parents reported a reflection or glare caused by lighting within the home when compared with children of parents that did not report a reflection or glare (Table 4, Figure A12). The severity of hyperactivity-impulsivity and combined ADHD were also higher in children whose parents reported too much daylight within their home

environment when compared to children whose parents did not report too much daylight (Table 4, Figures A13 and A14).

There was a borderline significant mean difference in the severity of combined ADHD in children whose parents reported insufficient artificial lighting within their home environment when compared with children of parents that did not report insufficient lighting (Table 4). This indicated that the severity of combined ADHD was higher in children living in a home environment with insufficient artificial lighting (Figure A15).

A significant association was found between the direction of the windows' placement within the home's main living area and the severity of hyperactivity-impulsivity ($p = 0.036$) and combined ADHD type ($p = 0.040$). The severity of hyperactivity-impulsivity and combined ADHD in children was higher when windows within the home's main living area faced south, being exposed to less sunlight or natural lighting. While children who lived in homes where the main windows faced towards the east or north, which received more sunlight or natural lighting, showed less severe symptoms (Table 4, Figures A16 and A17).

Furthermore, hyperactive-impulsive children and those with combined ADHD type experienced milder symptoms in homes where the main windows faced east, receiving morning light, while those living in homes where the main windows faced towards the west, indicative of receiving afternoon light, had more severe symptoms (Table 4, Figures A16 and A17).

### 3.4. ADHD and Thermal Comfort

The reported suitability of thermal comfort showed a very weak, negative, and significant relationship with the severity of combined ADHD in children, indicating that the severity of combined ADHD was significantly lower in children with a more suitable thermal comfort within their home (Table 5, Figure A18).

**Table 5.** Relationship between thermal comfort and severity of ADHD symptoms.

| Thermal Comfort | ADHD Severity | N | df | F | Pearson Correlation (2-Tailed) | Mean Difference | Std. Error | Sig. |
|---|---|---|---|---|---|---|---|---|
| Suitability of thermal comfort | Combined ADHD | 390 | - | - | −0.105 | - | - | 0.038 |
| Ceiling fan | Inattention | 390 | 1 | 5.19 | - | 0.227 | 0.100 | 0.023 |
|  | Combined ADHD | 390 | 1 | 5.63 | - | 0.261 | 0.110 | 0.018 |
| Permanent heater | Combined ADHD | 390 | 1 | 4.46 | - | 0.365 | 0.173 | 0.035 |
|  | Hyperactivity-impulsivity | 389 | 1 | 3.83 | - | 0.403 | 0.206 | 0.051 |
| Indoor gas fireplace | Combined ADHD | 386 | 1 | 5.41 | - | 0.417 | 0.179 | 0.021 |

Pairwise comparisons by Bonferroni.

There was a significant association between ADHD diagnosis and the use of a ceiling fan to control the air temperature (Fisher's Exact Test = 9.60, $p = 0.047$), indicating that children with the hyperactive-impulsive type (72.2%) were significantly more likely to use a ceiling fan within their home (Figure A19). Using a ceiling fan was also significantly correlated with the severity of inattention and combined ADHD type (Table 5). Children of parents who used a ceiling fan scored significantly lower on the severity of inattention and combined ADHD when compared to children of those who did not use a ceiling fan (Figures A20 and A21).

Also, using a permanent heater showed a significant correlation with the severity of combined ADHD type and hyperactivity-impulsivity (borderline significance) (Table 5). Children whose parents reported using a permanent heater scored lower on the severity of combined ADHD and hyperactivity-impulsivity than children whose parents did not

report using a permanent heater at home (Figures A22 and A23). Furthermore, a higher severity of combined ADHD type was found in children whose parents reported not having an indoor gas fireplace when compared with children with such a fireplace at home (Table 5, Figure A24).

Other thermal comfort issues within the home, such as having problems with tropical weather, having parts of the home very cold or very hot, or low airflow in some rooms, as reported by participants, also had a significant association with ADHD diagnosis (Fisher's Exact Test = 9.91, $p = 0.027$). Children with ADHD and comorbid conditions (88.2%) were more likely to report other thermal comfort issues in the home environment than children with combined ADHD type (97.4%) (Figure A25).

### 3.5. The Contribution of IEQ to the Prediction of ADHD

The multinomial logistic regression (with a 95% confidence interval) was conducted to determine the power of the indoor environmental quality in predicting an ADHD diagnosis and to explore the strongest contributing IEQ variable(s). The overall model of the contribution of the housing IEQ in the prediction of an ADHD diagnosis was significant ($\chi^2 (20) = 50.15$, $p < 0.001$). A 13.3% (Nagelkerke R-Square) of variance of having an ADHD diagnosis was explained by a collective contribution of the five variables, reflecting issues with the air quality, the suitability of acoustic quality, issues with the lighting quality, the controllability of thermal comfort, and the issues with thermal comfort within the home environment. Having an issue relating to the thermal comfort ($\chi^2 (4) = 9.91$, $p = 0.042$, $b = 1.62$) was the strongest predictor within the model, followed by artificial lighting flickers ($\chi^2 (4) = 10.02$, $p = 0.040$, $b = 1.18$) and using a ceiling fan ($\chi^2 (4) = 11.00$, $p = 0.027$, $b = -1.11$). While the suitability of acoustic quality ($\chi^2 (4) = 11.79$, $p = 0.019$, $b = 0.57$) was indicated to be the weakest predictor within the model.

A Stepwise linear regression was performed to measure the power of the indoor environmental quality in predicting ADHD severity and to explore the strongest contributing IEQ variable(s). A significant collective correlation was found between the indoor environmental quality and the severity of all the ADHD symptoms ($p < 0.001$). Air quality and thermal comfort predicted 4% of the variance in severity of inattention ($F (3, 383) = 5.32$, $R^2 = 0.040$). Using a vacuum cleaner was the strongest predictor ($t = -2.32$, $p = 0.021$, $\beta = -0.53$), and using a ceiling fan was the weakest predictor ($t = -2.30$, $p = 0.022$, $\beta = -0.22$) in this model.

The air and lighting quality predicted 9.7% of the variance in severity of hyperactivity-impulsivity ($F (5, 359) = 7.73$, $R^2 = 0.097$). Using a vacuum cleaner ($t = -3.91$, $p < 0.001$, $\beta = -1.17$) was the strongest predictor in the model, followed by reflection or glare caused by lighting ($t = 2.62$, $p = 0.009$, $\beta = 0.50$). Windows of the main living area facing north ($t = -2.09$, $p = 0.037$, $\beta = -0.30$) and too much daylight ($t = 1.90$, $p = 0.059$, $\beta = 0.39$) were the weakest predictors in this model.

The air quality, acoustic quality, lighting quality, and thermal comfort predicted 10.9% of the variance in severity of combined ADHD ($F (5, 369) = 9.06$, $R^2 = 0.109$). Using a vacuum cleaner ($t = -4.50$, $p < 0.001$, $\beta = -1.13$) was the strongest predictor in the model, followed by windows of the main living area facing east ($t = -3.23$, $p = 0.001$, $\beta = -0.46$). The suitability of acoustic quality ($t = -2.21$, $p = 0.028$, $\beta = -0.12$) and using a ceiling fan ($t = -2.14$, $p = 0.033$, $\beta = -0.22$) were the weakest predictors in this model.

## 4. Discussion

Children spend most of their time indoors, with most of the existing housing being designed and built without considering their needs, especially for those with different cognitive abilities, such as children diagnosed with ADHD who are extra sensitive to the impacts of their everyday surroundings. Therefore, this study aimed to identify any housing indoor environmental qualities that may adversely impact children's ADHD symptoms and behaviours.

Our findings build on the literature regarding the environmental influences on ADHD in other settings, such as schools (e.g., the impact of noise on children with ADHD in a study conducted by Tufvesson [61]). The results of our study showed a relationship between all the IEQ factors with ADHD diagnosis and the severity of the symptoms. More specifically, the housing indoor environmental quality was significantly associated with a diagnosis of ADHD relative to the issues of air quality, the suitability of acoustic quality, issues with lighting quality, the controllability of thermal comfort, and issues with thermal comfort. Among these items, having an issue with the thermal comfort was the strongest contributor, followed by issues associated with the lighting quality and the controllability of thermal comfort.

Also, the housing IEQ was significantly associated with the severity of ADHD symptoms in terms of the controllability of air quality, issues with air quality, suitability of acoustic quality, the direction of the main home windows, the suitability of artificial lighting, issues with lighting quality, the controllability of thermal comfort, and the suitability of thermal comfort. The controllability of air quality was the strongest contributor, followed by the main home windows' direction and the thermal comfort's controllability.

Some of our results were congruent with prior studies, such as the findings about the importance of air, lighting, and the acoustic quality for children with ADHD. In support of our findings, the previous research has linked the development of neurological disorders, learning disabilities, and cognitive deficits (such as ADHD and ASD) with exposure to environmental pollutants and toxic chemicals [44–46].

Consistent with our findings, studies have also shown that artificial light [66] and daylight [68] affect children's behaviour and concentration ability. Although natural lighting is one of the important elements to achieve adequate lighting in the area for ADHD and ASD [82], as also shown in our findings, too much daylight or direct natural lighting (glare) can increase distraction and affect a child's performance, especially at school [62,76,77].

In line with our findings, the previous research has also shown that low acoustic quality or noise adversely influenced children's psychological wellbeing [54,55] and caused inattention and misbehaviour in children with ADHD and ASD [64,76,82].

Some of the research outcomes are novel and is considered to be an initial step in making a significant contribution to the existing knowledge. In this regard, our research revealed that maintaining thermal comfort in the home environment is more significant for children with ADHD, which has been poorly covered in the literature, if not neglected. Thus, our findings suggest that besides having a suitable acoustic quality, air (such as air free from unpleasant smells or dust), and sufficient lighting (avoiding too much daylight or insufficient artificial lighting), it is also important to use a heater and fan to provide a comfortable temperature within the home environment. Morning daylight and using a vacuum cleaner for cleaning the home were also found to be important.

One of the limitations is that, although children and adolescents in this study were diagnosed to have ADHD by a healthcare professional, there may be a measurement bias as we relied only on the parental report of the ADHD symptoms and severity.

Despite this limitation, the current study is among the first nationally representative studies with a large sample size to document the associations between ADHD among children and adolescents with their home IEQ in Australia.

## 5. Conclusions

In conclusion, this research has provided insights into the importance of housing features and suggests that improving the housing indoor environmental quality, mainly thermal comfort, air, and lighting quality could positively correlate with alleviating ADHD symptoms and severity among children and adolescents. While the association between the IEQ and ADHD diagnosis/severity does not equate to causality, the findings suggest that the IEQ is an important determinant in creating an ADHD-friendly living space.

Future research should extend the findings of this exploratory research by qualitatively testing the associations between the housing indoor environmental quality and ADHD

with a focused sample group as well as qualitatively and quantitatively testing the housing preferences by children themselves rather than their parents.

**Author Contributions:** Conceptualization, S.A.; Data curation, S.A.; Funding acquisition, S.A.; Investigation, S.A.; Methodology, S.A.; Supervision, C.E.B., B.H.J. and V.E.; Validation, S.A.; Writing—original draft, S.A.; Writing—review & editing, C.E.B., B.H.J. and V.E. All authors have read and agreed to the published version of the manuscript.

**Funding:** This research was funded by an Australian Government Research Training Program Scholarship.

**Institutional Review Board Statement:** The study was approved by the Institutional Ethics Committee of the University of New South Wales (HC200539; 06/10/2020).

**Informed Consent Statement:** Informed consent was obtained from all subjects involved in the study.

**Data Availability Statement:** Not applicable.

**Acknowledgments:** The authors would like to acknowledge the Commonwealth's support and fund allocated to this project by an "Australian Government Research Training Program Scholarship". We are also grateful to those who participated in this research. This work is part of a research thesis in partial fulfilment of the degree of Doctor of Philosophy at the University of New South Wales, Australia.

**Conflicts of Interest:** The authors declare no conflict of interest.

## Appendix A. ADHD and Air Quality

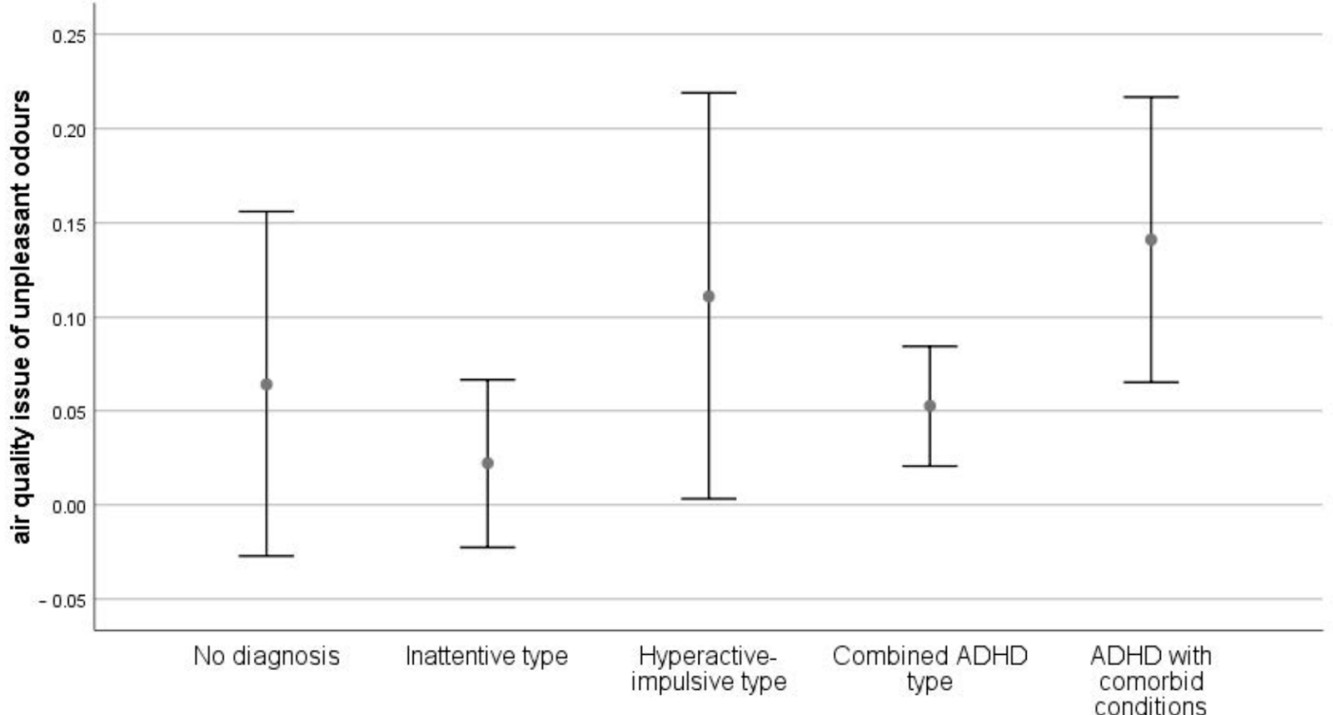

**Figure A1.** Significant difference in the diagnosis of ADHD symptoms in terms of air quality issues within the home: unpleasant odours.

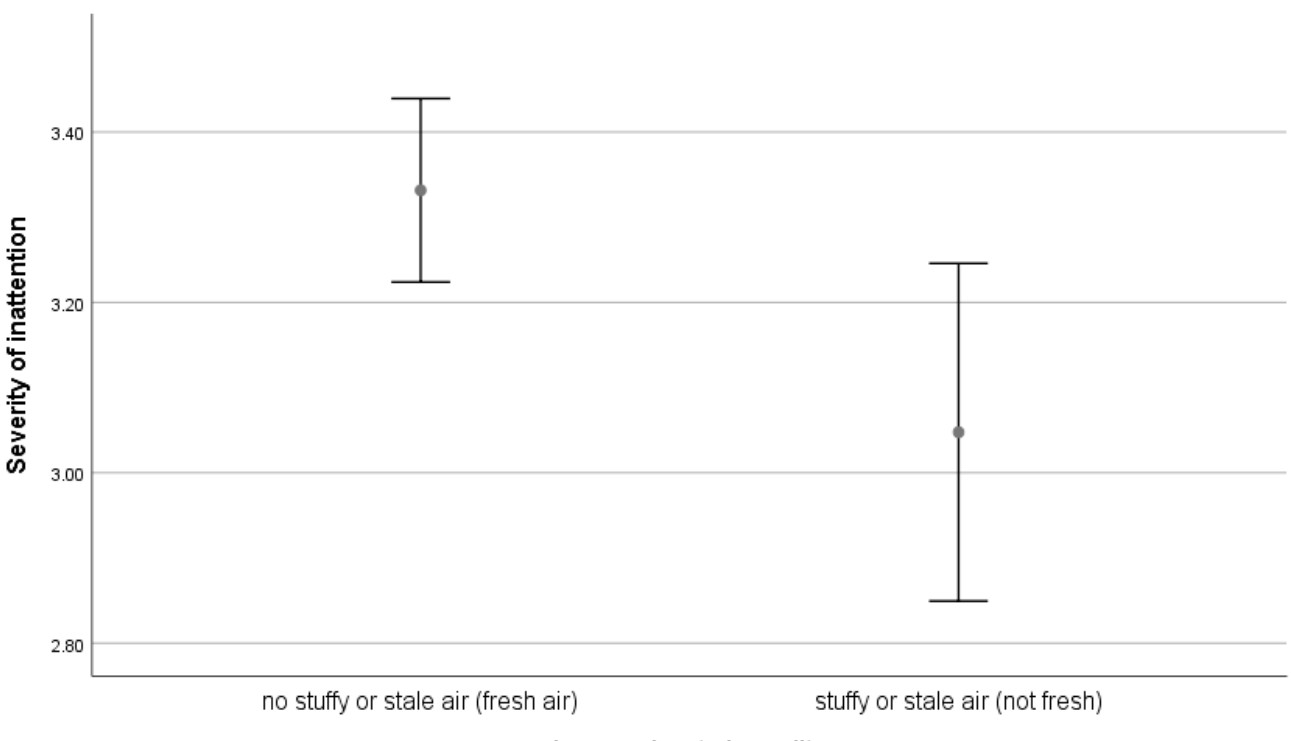

**Figure A2.** Significant difference in the severity of inattention in terms of issues about air quality in the home: stuffy or stale air.

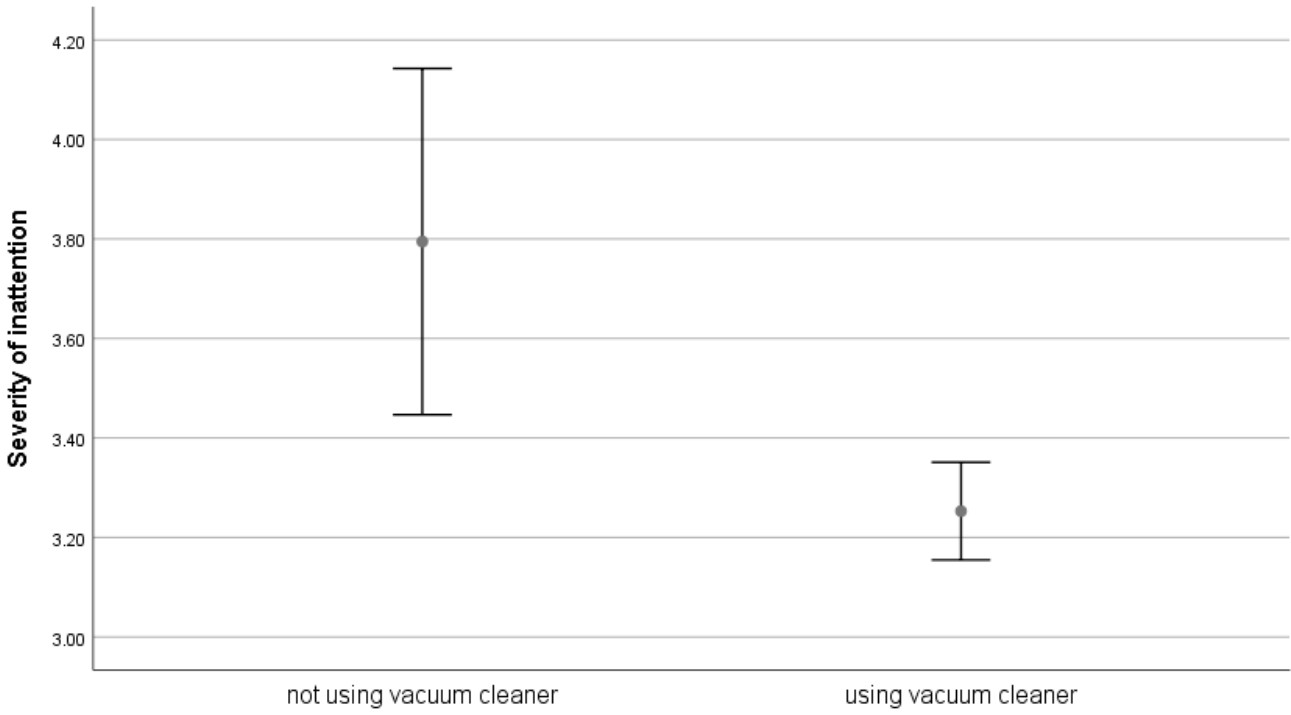

**Figure A3.** Significant difference in the severity of inattention in terms of using a vacuum cleaner to control air quality within the home.

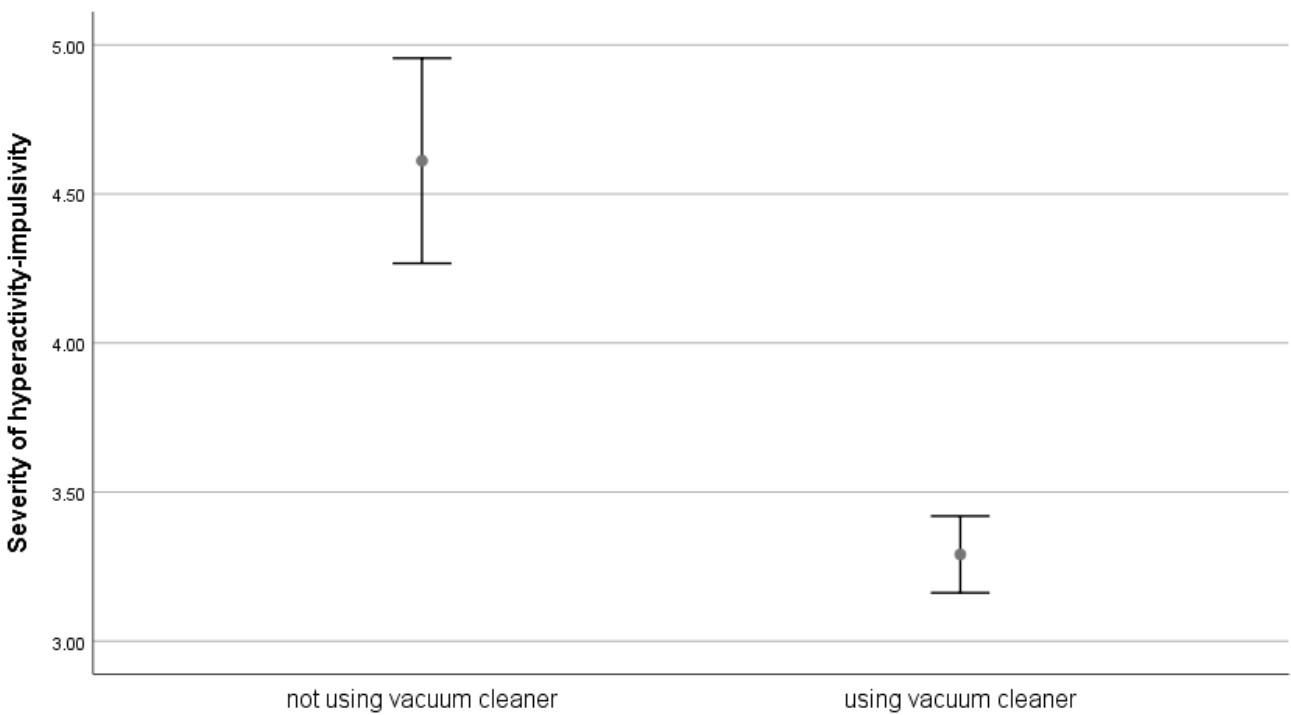

**Figure A4.** Significant difference in the severity of hyperactivity-impulsivity in terms of using a vacuum cleaner to control air quality within the home.

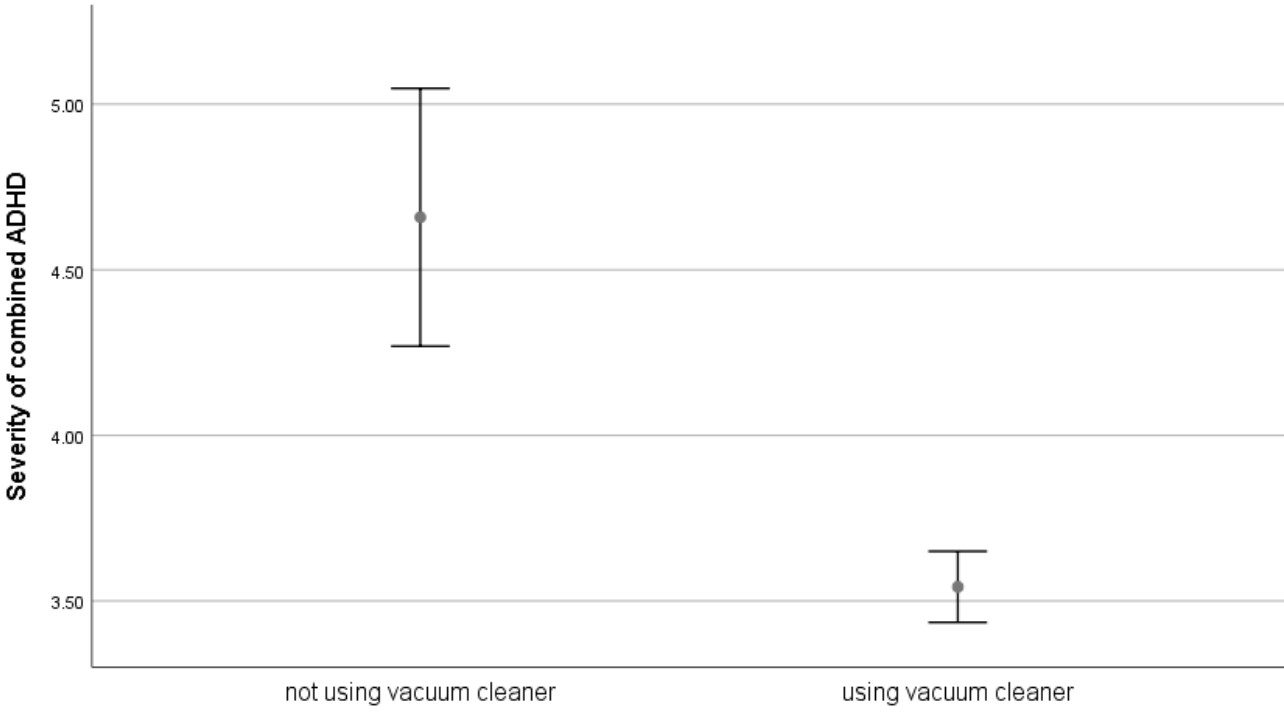

**Figure A5.** Significant difference in the severity of combined ADHD in terms of using a vacuum cleaner to control air quality within the home.

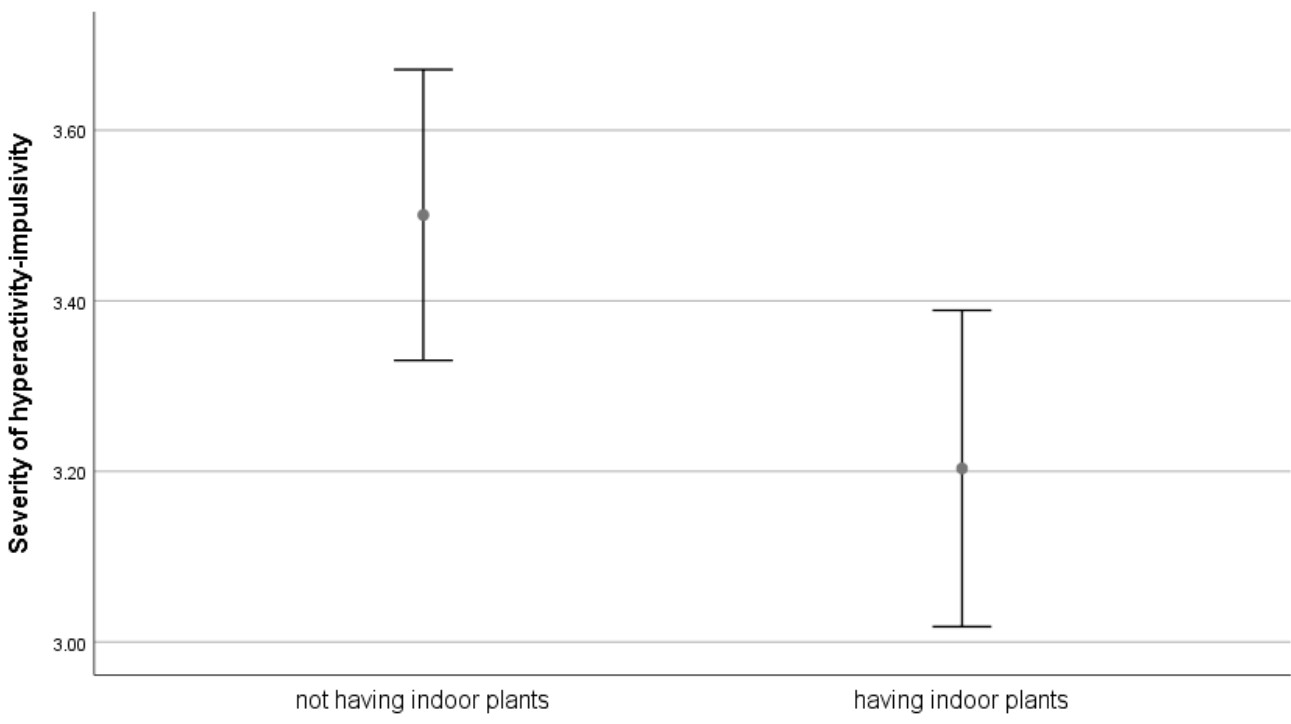

**Figure A6.** Significant difference in the severity of hyperactivity-impulsivity in terms of having indoor plants to adjust air quality within the home.

## Appendix B. ADHD and Acoustic Quality

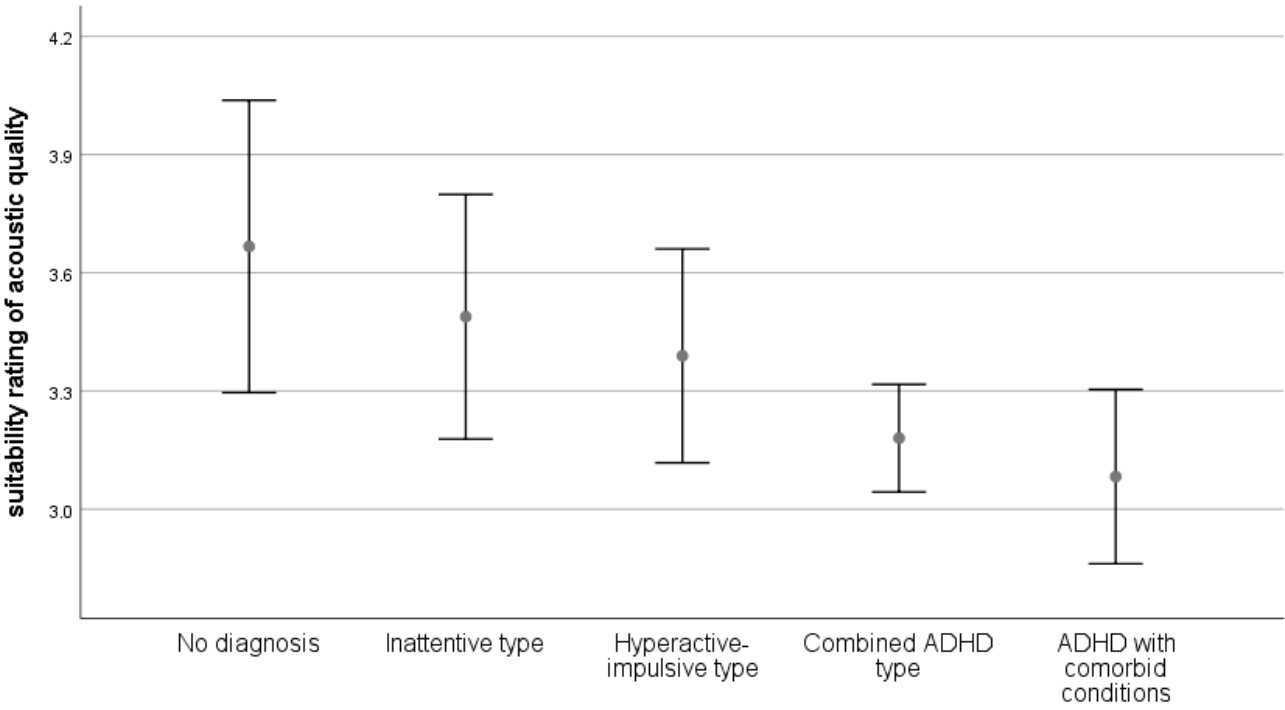

**Figure A7.** Significant difference in the diagnosis of ADHD symptoms in terms of the suitability rating of acoustic quality.

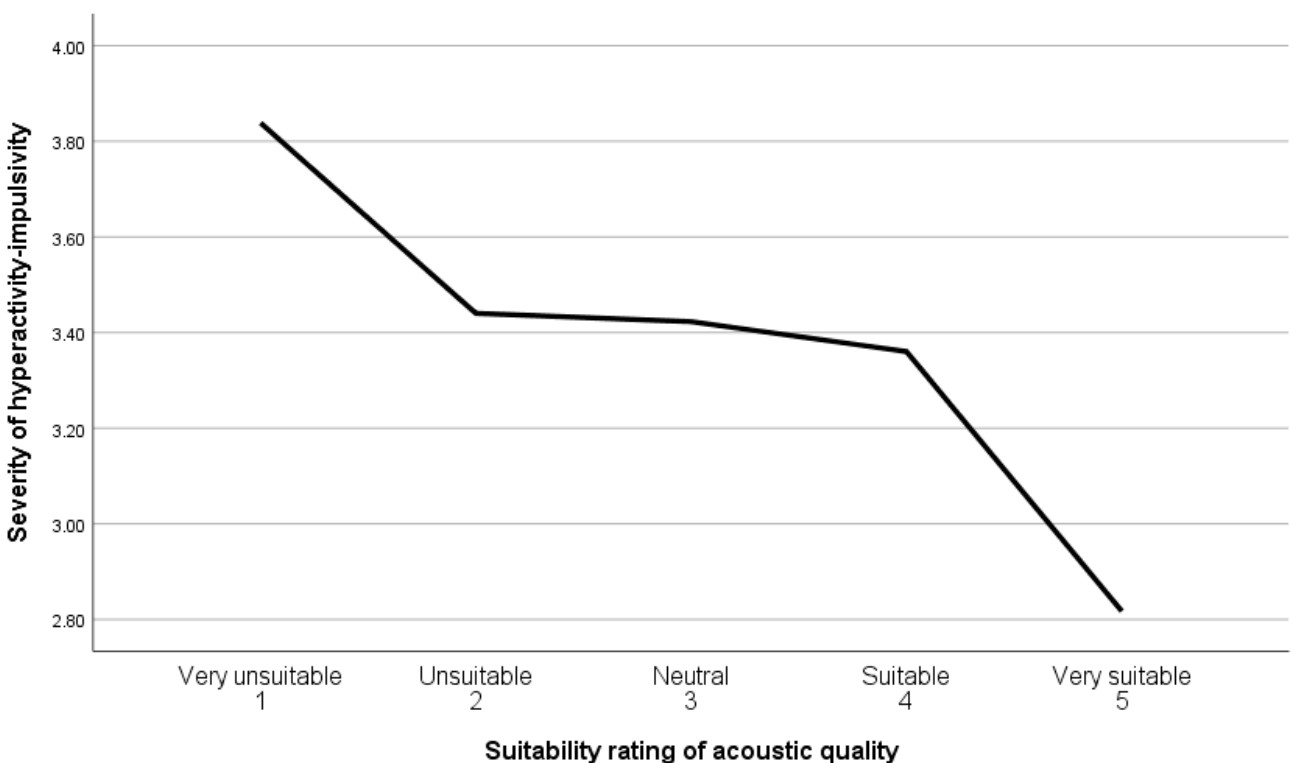

**Figure A8.** Significant relationship between the severity of hyperactivity-impulsivity and the suitability rating of acoustic quality within the home.

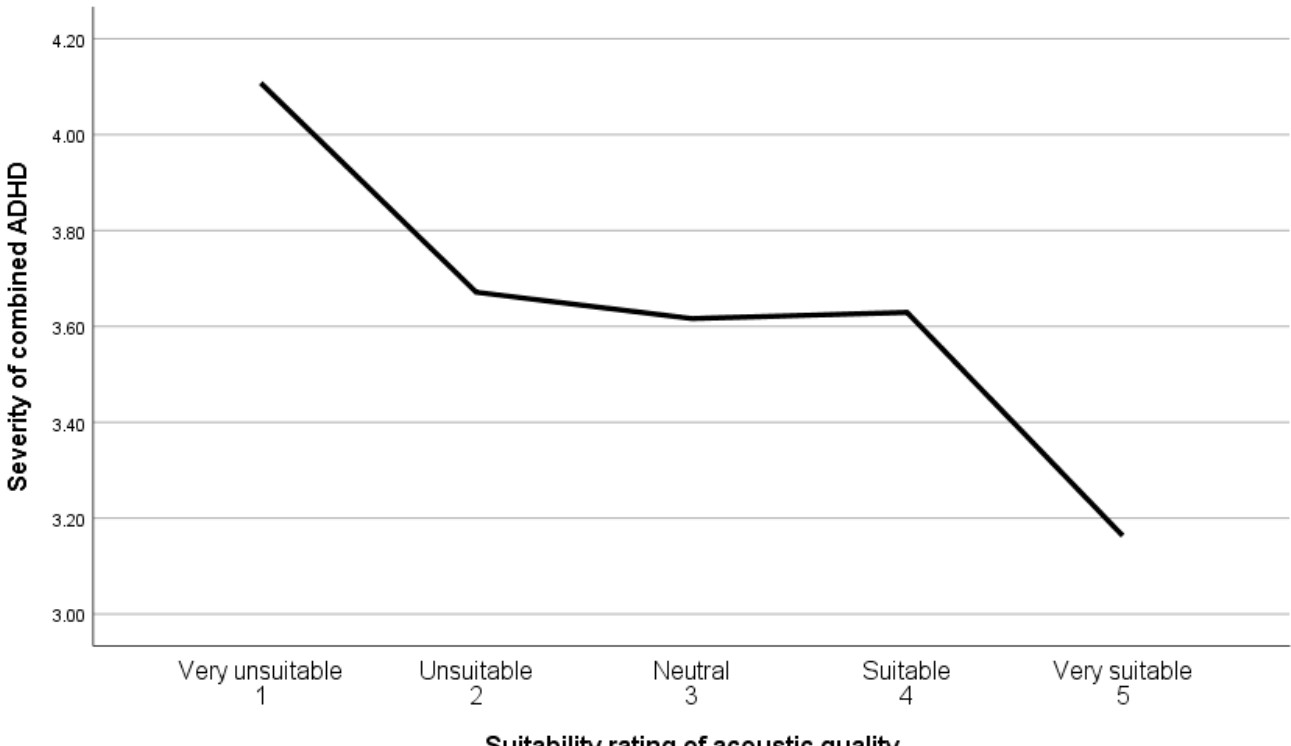

**Figure A9.** Significant relationship between the severity of combined ADHD and the suitability rating of acoustic quality within the home.

## Appendix C. ADHD and Lighting Quality

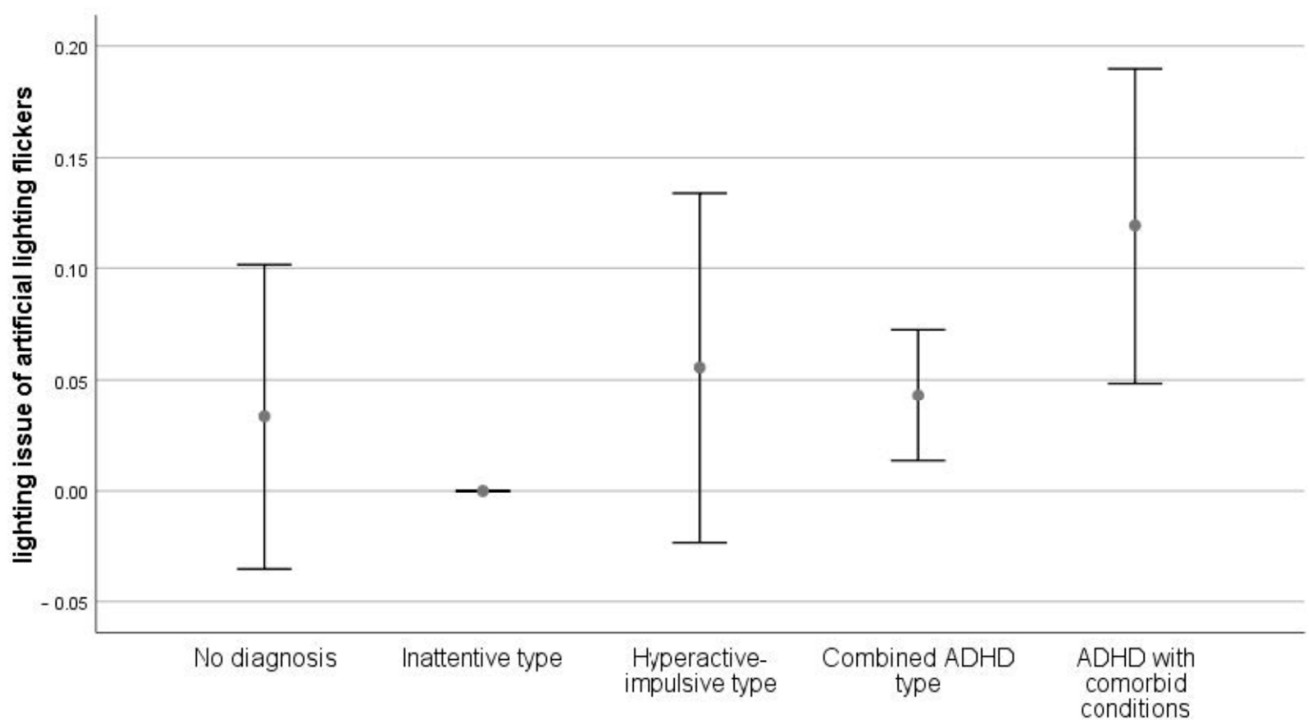

**Figure A10.** Significant difference in the diagnosis of ADHD symptoms in terms of lighting quality issues within the home: artificial lights flickering.

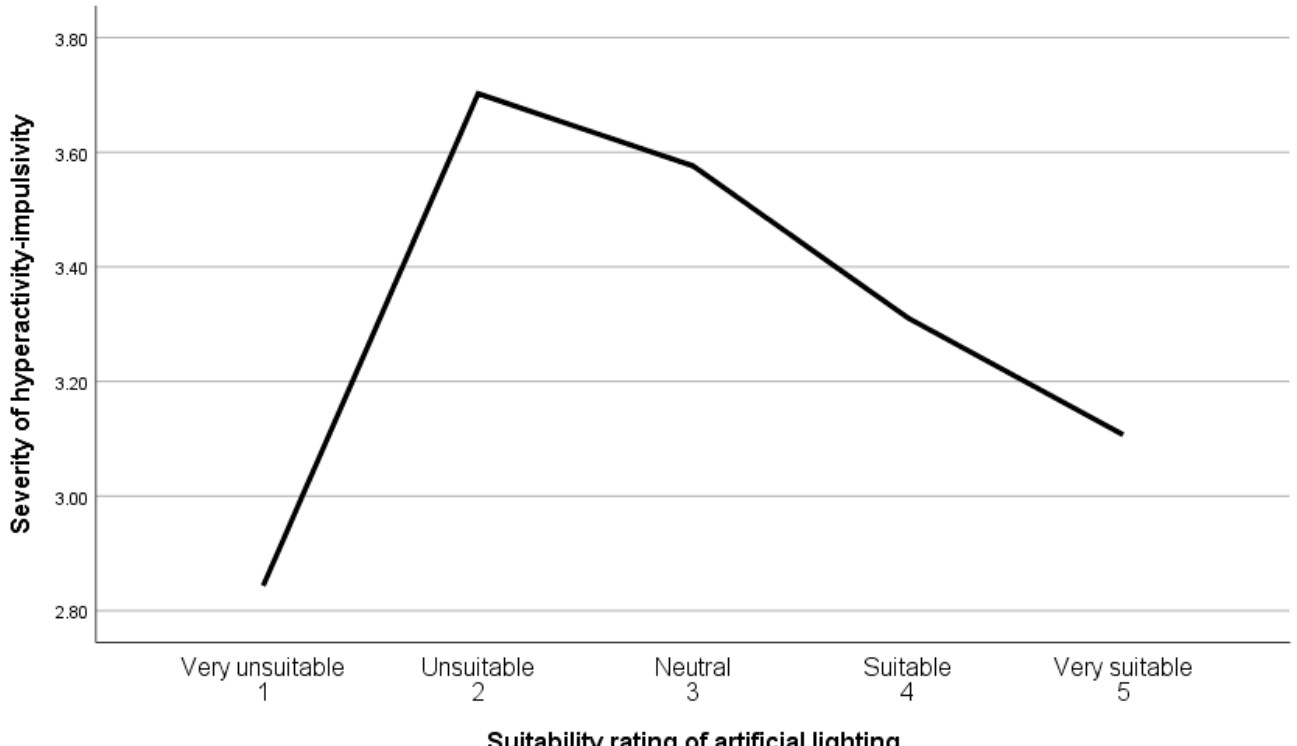

**Figure A11.** Significant relationship between the severity of hyperactivity-impulsivity and the suitability rating of artificial lighting within the home.

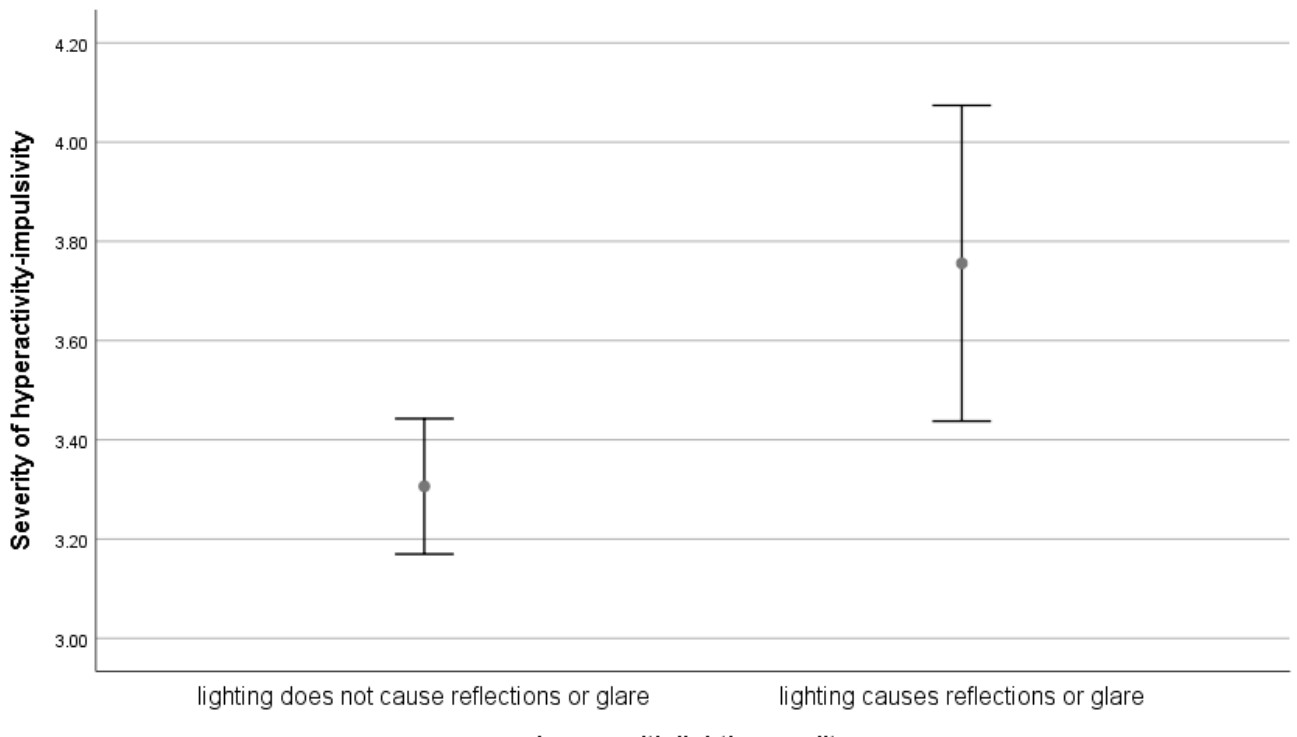

**Figure A12.** Significant difference in the severity of hyperactivity-impulsivity in terms of issues with lighting quality within the home: reflections or glare caused by lighting.

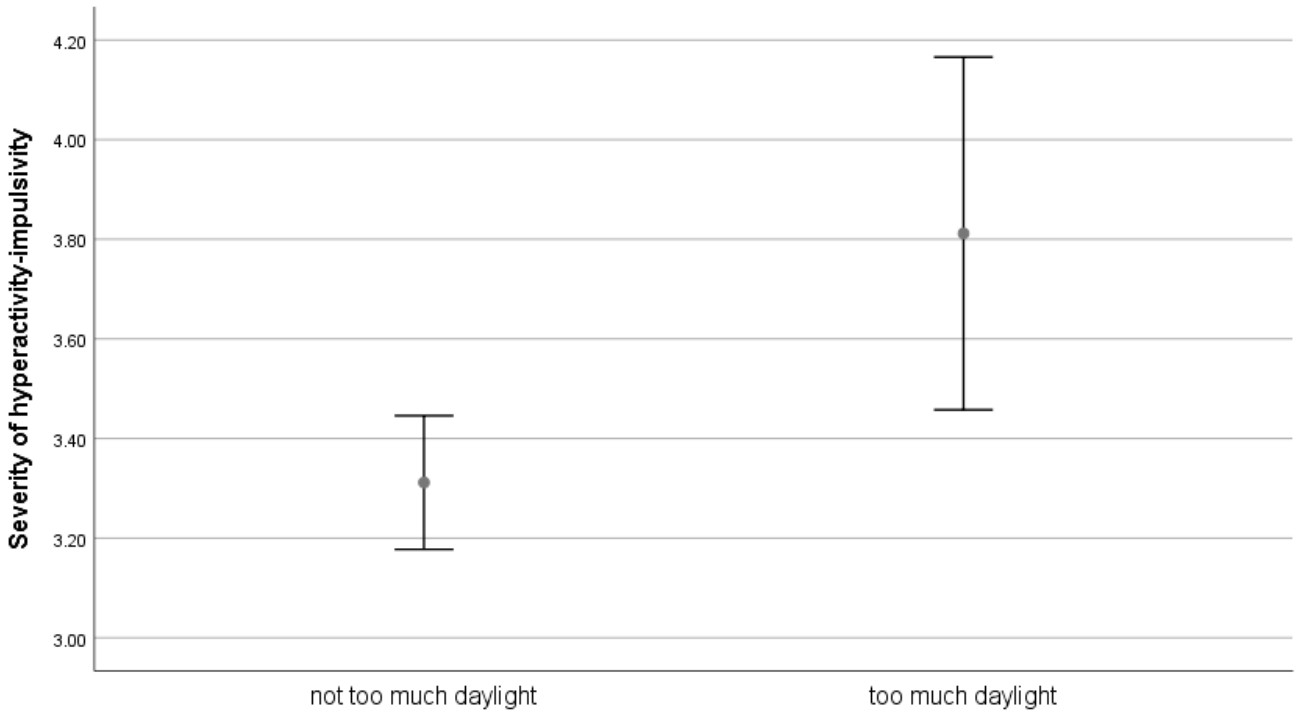

**Figure A13.** Significant difference in the severity of hyperactivity-impulsivity in terms of issues with lighting quality within the home: too much daylight.

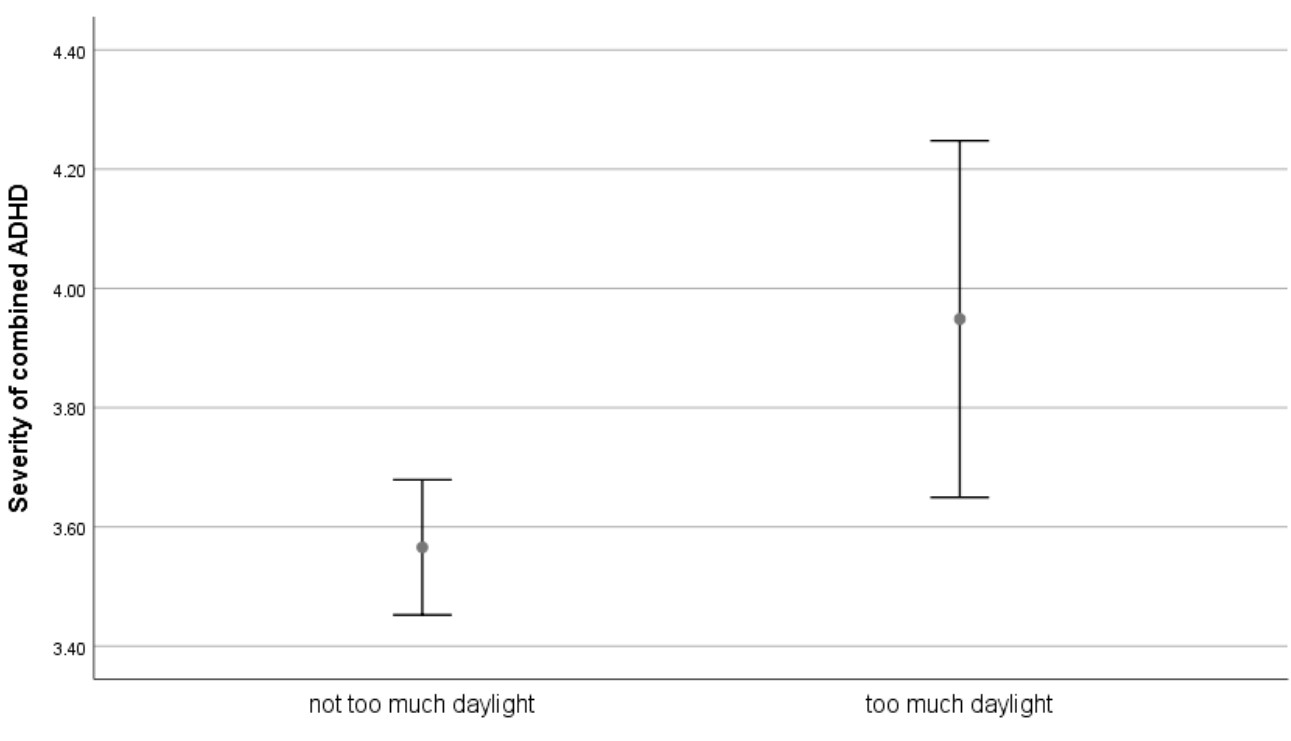

**Figure A14.** Significant difference in the severity of combined ADHD in terms of issues with lighting quality within the home: too much daylight.

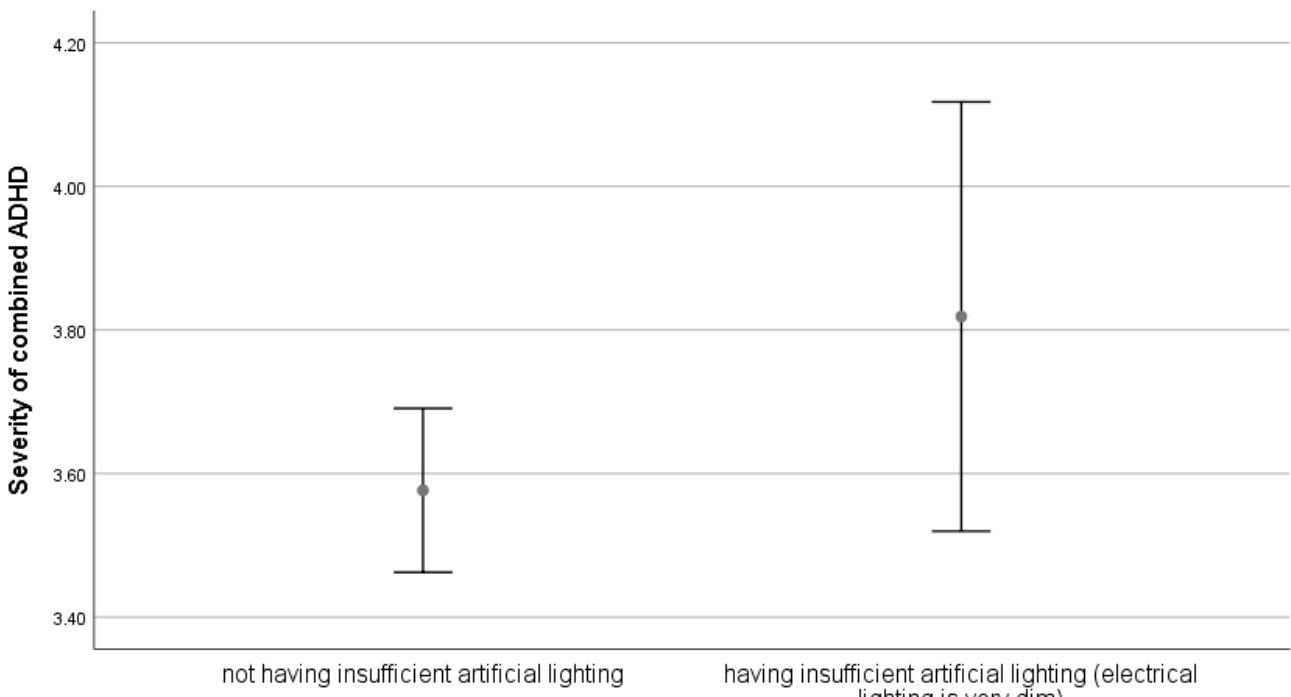

**Figure A15.** Significant difference in the severity of combined ADHD in terms of issues about lighting quality within the home: insufficient artificial lighting.

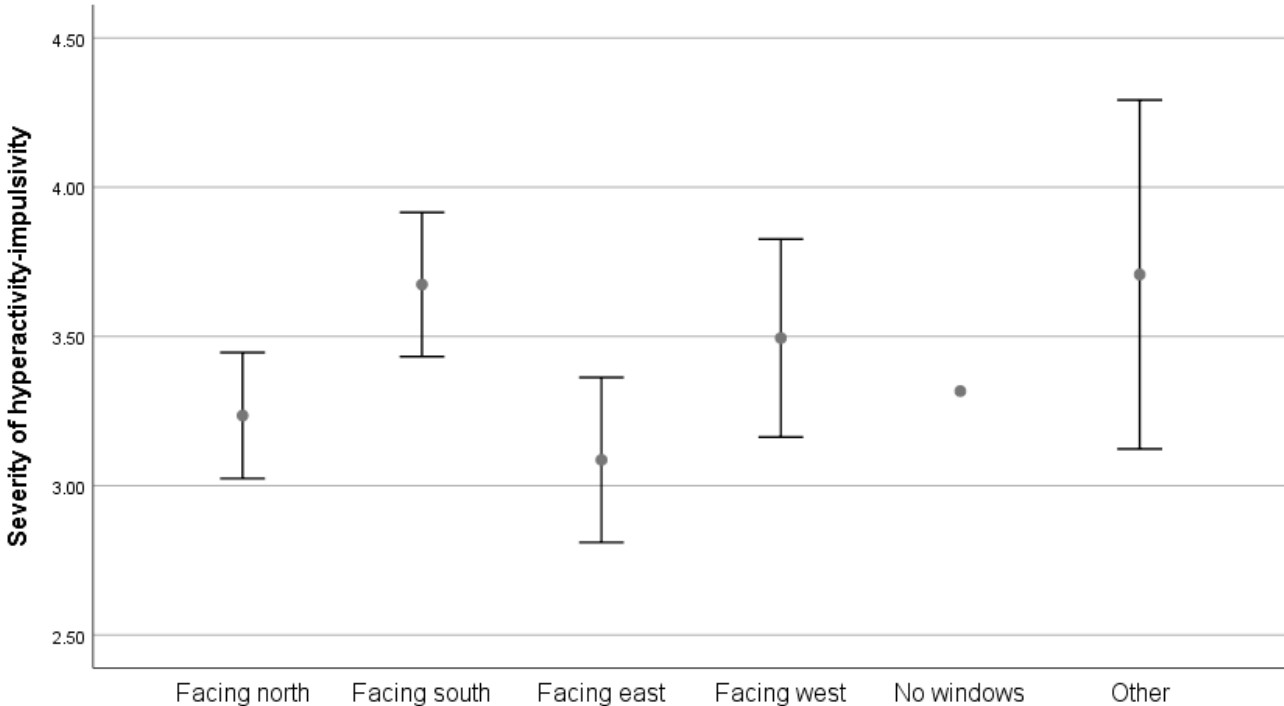

**Figure A16.** Significant difference in the severity of hyperactivity-impulsivity in terms of the direction of home main windows: amounts of sunlight exposure in the home.

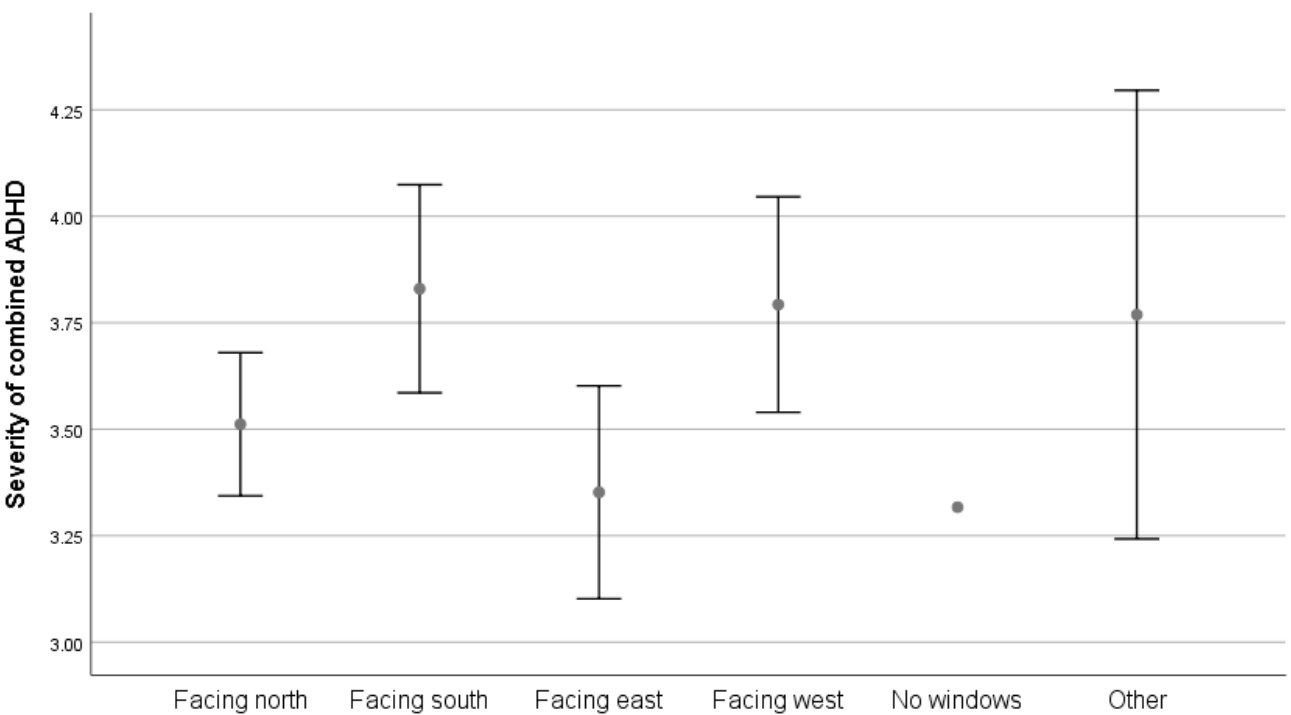

**Figure A17.** Significant difference in the severity of combined ADHD in terms of the direction of home main windows: the amount of sunlight exposure in the home.

**Appendix D. ADHD and Thermal Comfort**

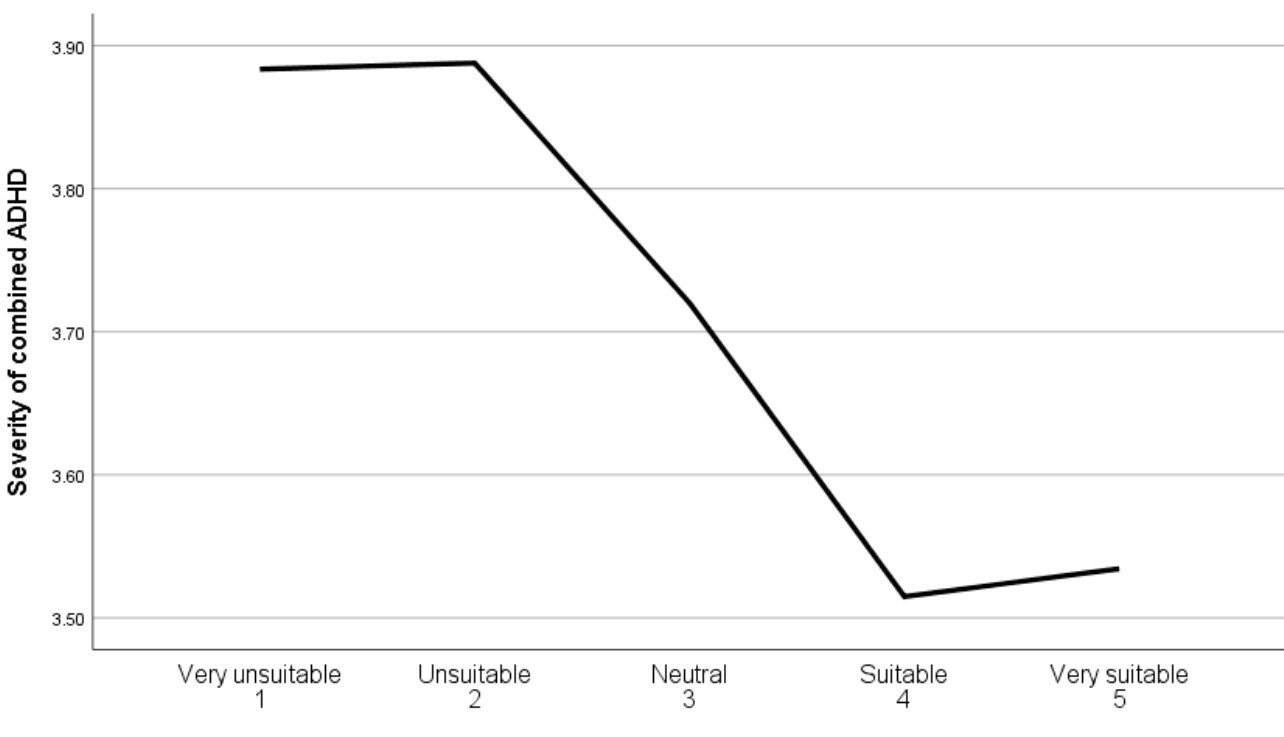

**Figure A18.** Significant relationship between the severity of combined ADHD and the suitability rating of thermal comfort within the home.

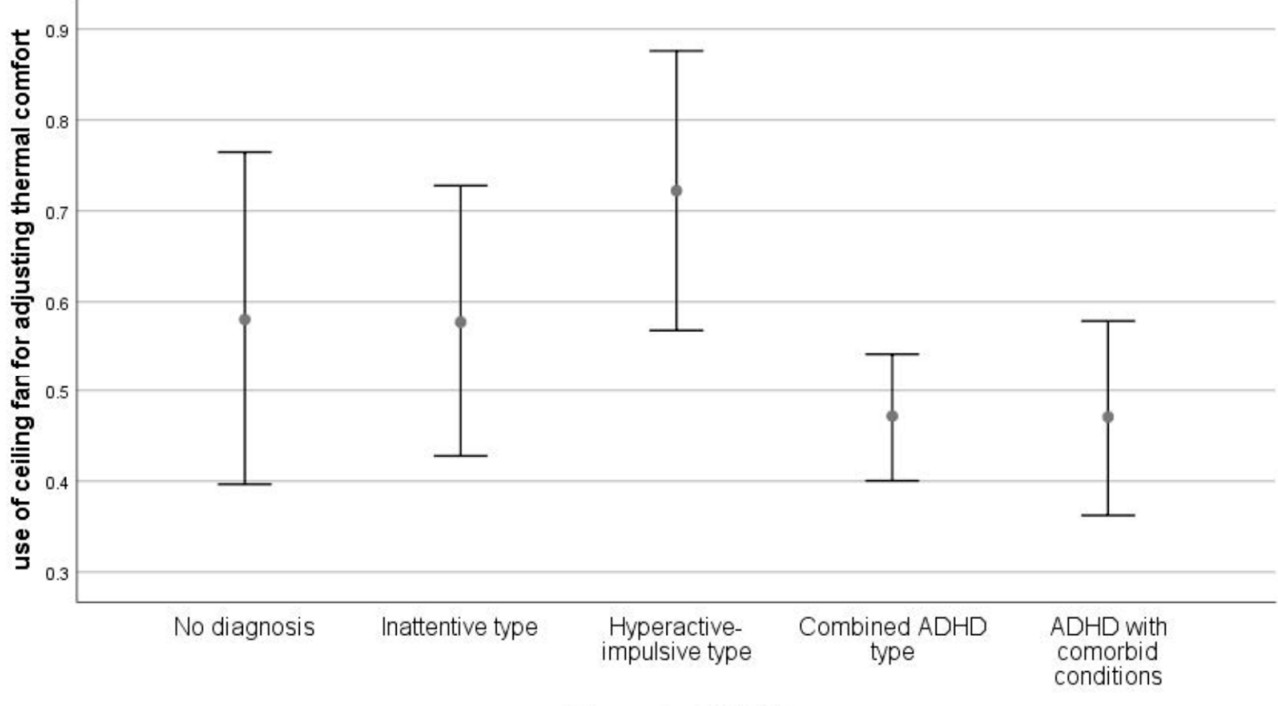

**Figure A19.** Significant difference in the diagnosis of ADHD symptoms in terms of using a ceiling fan to adjust or control thermal comfort within the home.

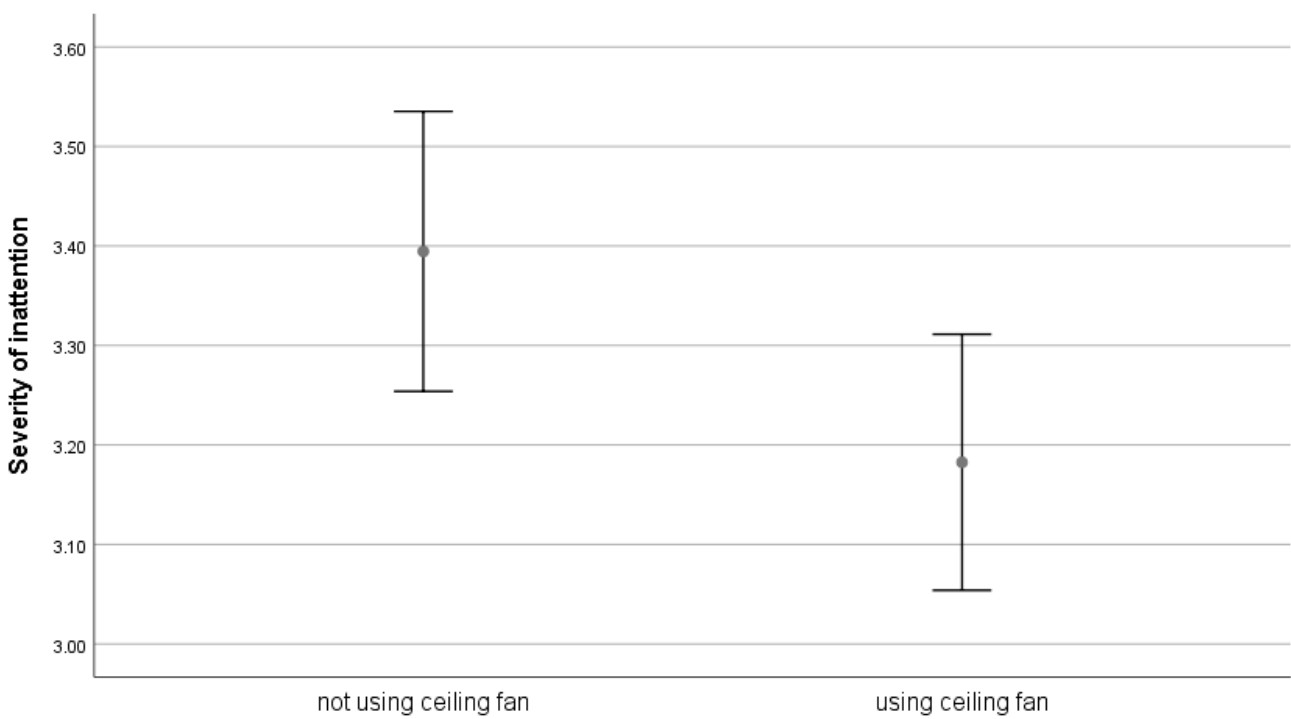

**Figure A20.** Significant difference in the severity of inattention in terms of using a ceiling fan to adjust or control thermal comfort within the home.

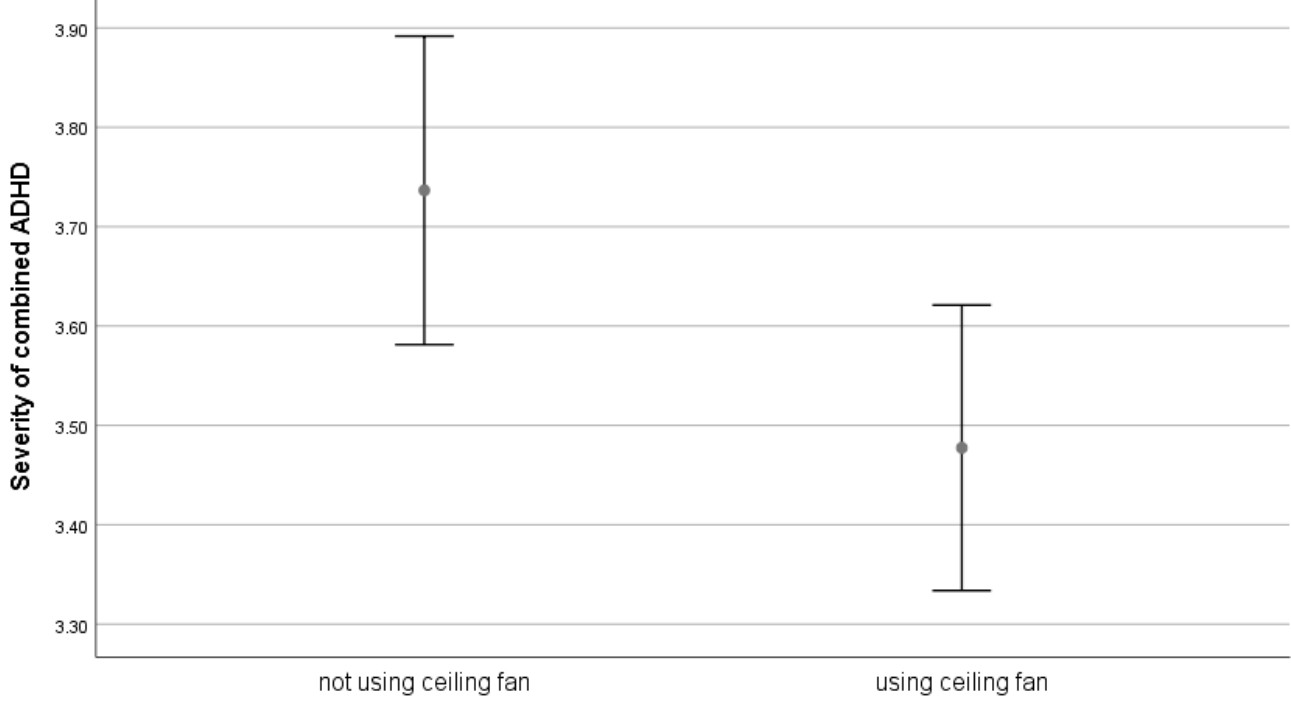

**Figure A21.** Significant difference in the severity of combined ADHD in terms of using a ceiling fan to adjust or control thermal comfort within the home.

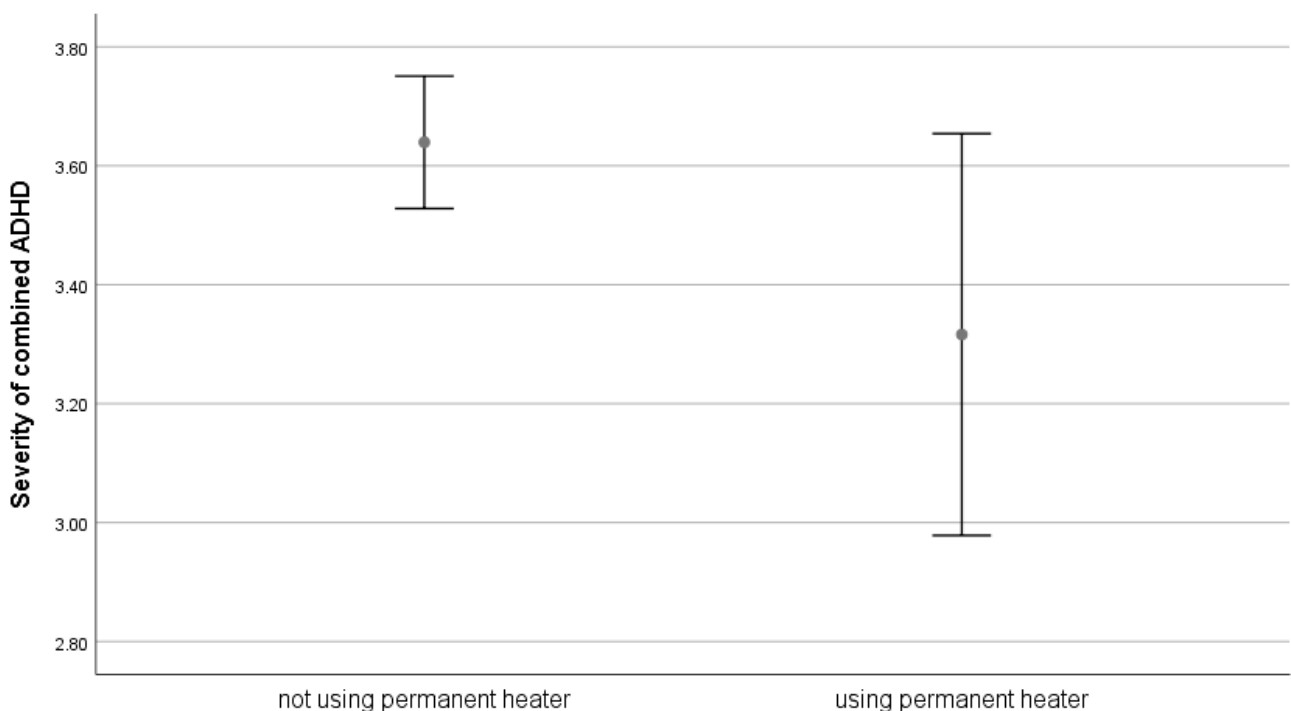

**Figure A22.** Significant difference in the severity of combined ADHD in terms of using a permanent heater to adjust or control thermal comfort within the home.

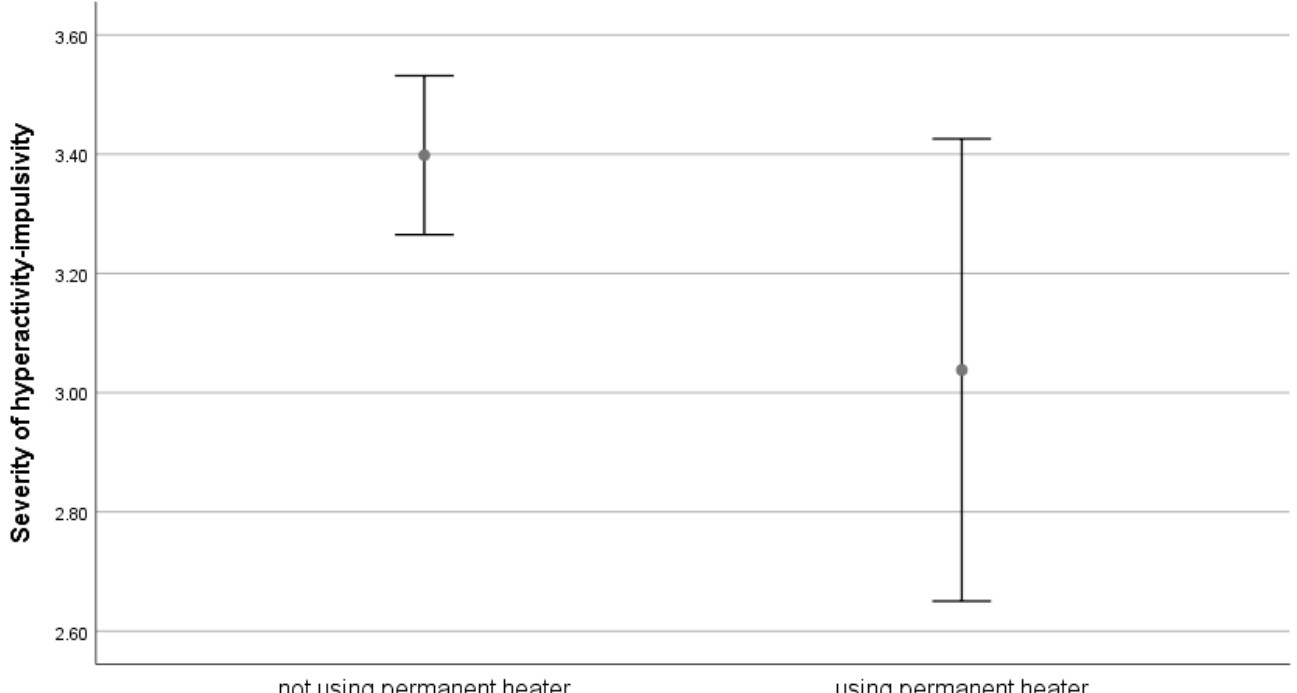

**Figure A23.** Significant difference in the severity of hyperactivity-impulsivity in terms of using a permanent heater to adjust or control thermal comfort within the home.

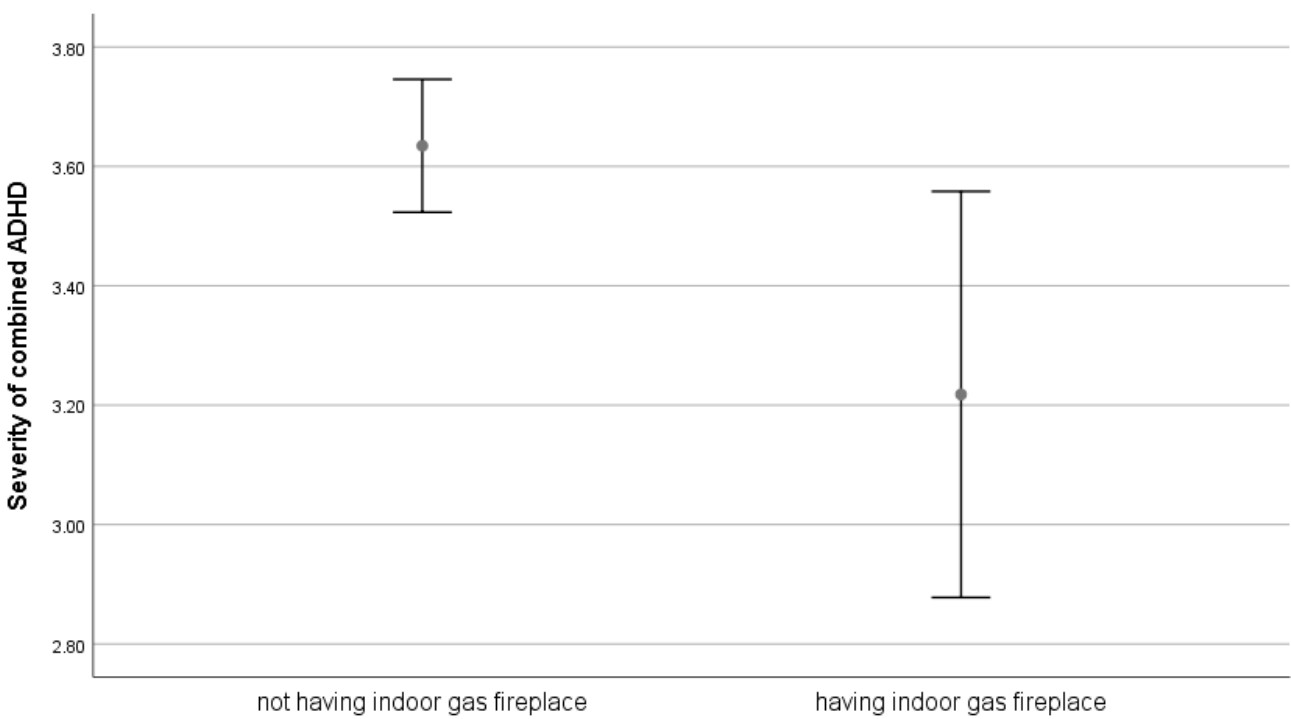

**Figure A24.** Significant difference in the severity of combined ADHD in terms of having an indoor gas fireplace within the home.

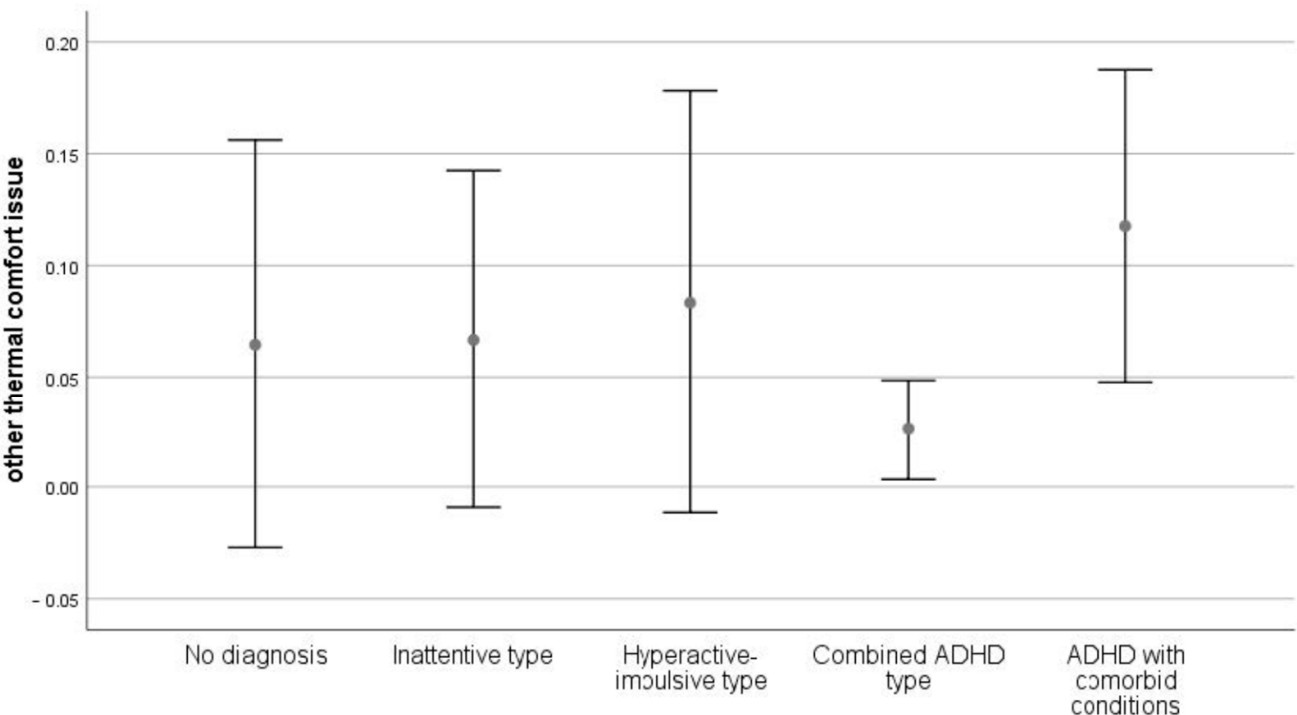

**Figure A25.** Significant difference in the diagnosis of ADHD symptoms in terms of other thermal comfort issues within the home.

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
