# Peer review of "Home Indoor Environmental Quality and Attention Deficit Hyperactivity Disorder"

_sustainability, doi:10.3390/su15042899_

Round 1
Reviewer 1 Report
I have carefully reviewed the paper entitled "Home Indoor Environmental Quality and ADHD”. To start with, I am satisfied that the manuscript contains original material and has the potential to make an interesting contribution in the field. However, before I recommend its publication, the authors should carry out the following revisions:
· Authors should list the software they used to process the data.
· More information should be provided on the measured parameters regarding the indoor air quality, thermal comfort, lighting quality and acoustic quality of the spaces in which the questionnaires were conducted. Tables should be given with the statistics (mean, standard deviation) of all measured parameters.
· The novelty of the manuscript should be highlighted better. What are the key new insights that this paper reports? Likewise, the abstract and conclusions section should be improved.
Author Response
Dear Reviewer,
Thank you very much for your helpful and constructive comments. Please find my point-by-point response to your comments below and the revised manuscript (Track Changes version) in the attachment. Please let me know whether more information is needed.
Response to Reviewer 1 Comments
Point 1: Authors should list the software they used to process the data.
Response 1: The research analyses were carried out using IBM SPSS Statistics software. (Page 4, Line 184)
Point 2: More information should be provided on the measured parameters regarding the indoor air quality, thermal comfort, lighting quality and acoustic quality of the spaces in which the questionnaires were conducted. Tables should be given with the statistics (mean, standard deviation) of all measured parameters.
Response 2: Housing IEQ was measured by both objective and subjective format questions relating to four aspects: acoustic quality, air quality, lighting quality, and thermal comfort. Subjective-format questions such as the rate of suitability were measured using 5-point Likert scales with the choices from very unsuitable to very suitable. Objective-format questions such as level of controllability or relevant issues were measured using multiple-choice questions and required participants to identify attributes or conditions at home such as the presence or absence of specific items. However, the studied housing IEQ parameters were not measured on the site by the researchers as the survey was conducted in the form of an anonymous (self-administered) online questionnaire. (P4-5, L194 & 199-208)
- A table representing the statistics of all parameters is included. (Table 1, P5, L209-213)
Point 3: The novelty of the manuscript should be highlighted better. What are the key new insights that this paper reports? Likewise, the abstract and conclusions section should be improved.
Response 3:
- The novelty of the manuscript has been further highlighted as follows:
Some of our results were congruent with prior studies such as the findings about the importance of air, lighting, and acoustic quality for children with ADHD. In support of our findings, previous research has linked the development of neurological disorders, learning disabilities, and cognitive deficits (such as ADHD and ASD) with exposure to environmental pollutants and toxic chemicals [45-47].
Consistent with our findings, studies have also shown that artificial light [67] and daylight [69] affect children’s behaviour and concentration ability. Although natural lighting is one of the important elements to achieve adequate lighting in the area for ADHD and ASD [83], as also shown in our findings, too much daylight or direct natural lighting (glare) can increase distraction and affect a child’s performance, especially at school [63, 77, 78].
In line with our findings, former research has also shown that low acoustic quality and noise adversely influenced children’s psychological wellbeing [55, 56] and caused inattention and misbehaviour in children with ADHD and ASD [65, 77, 83].
Some of the research outcomes are novel and considered as an initial step that makes a significant contribution to existing knowledge. In this regard, our research revealed that maintaining thermal comfort in the home environment is more significant for children with ADHD which has been poorly covered in the literature, if not neglected. Thus, our findings suggest that besides having suitable acoustic quality, air (such as air free from unpleasant smell or dust), and sufficient lighting (avoiding too much daylight or insufficient artificial lighting), it is important to use a heater and fan to provide a comfortable temperature within the home environment. Morning daylight and using a vacuum cleaner for cleaning the home were also found important. (P9, L364-386)
- The abstract section has been updated. (P1, L25-29)
- The conclusion section has been improved. (P10, L398-407)

Reviewer 2 Report
The proposed manuscript deals with home indoor environmental quality and the attention deficit hyperacivity disorder. The experimental part well designed and implemended using the meuserements of chemical and physical factors related to home indoor environmental quality.Statistics and data analysis well performed usinig proper statitistical analysis .
I propose to accept for publication to Sustainability taking into consideration the remarks:
Title ...and ADHD vs Attention Deficit Hyperactivity Disrder.
Page 2,line 64, 2. Background can be incorporated into intoduction
page 5,line 198 ,4.2 must moved in the begining, instead 4.1.
page 8,line 311,the discussion can be increased and the conclutions should be dicreased.
Author Response
Dear Reviewer,
Thank you very much for your helpful and constructive comments. Please find my point-by-point response to your comments below and the revised manuscript (Track Changes version) in the attachment. Please let me know whether more information is needed.
Response to Reviewer 2 Comments
Point 1: Title ...and ADHD vs Attention Deficit Hyperactivity Disrder [sic].
Response 1: Home Indoor Environmental Quality and Attention Deficit Hyperactivity Disorder (Page 1, Line 2-3)
Point 2: Page 2, line 64, 2. Background can be incorporated into intoduction [sic]
Response 2: Done (P2, L67)
Point 3: page 5, line 198, 4.2 must moved in the begining [sic], instead 4.1.
Response 3: Done (P5-6, L214-242)
Point 4: page 8, line 311, the discussion can be increased and the conclutions [sic] should be dicreased [sic].
Response 4: The discussion has been increased (P9, L343-348 & 364-386) and the conclusion has been decreased. (P9-10, L387-417)

Reviewer 3 Report
(1) The logic of the paper has to be improved. The introduction and background parts should be combined, and the section on IEQ's influence should be somewhat condensed (introduction section). It is necessary to link information on behavioral toxicity, smoking, carbon dioxide, and other topics with cues and, ideally, to see a clearer contextual framework (e.g., the impact of acoustic environmental quality, air quality, lighting quality, and thermal comfort on mental health). The research state and current issues should be summed up at the end.
(2) There is no data analysis in place. Tables 1, 2, and 3's data haven't been thoroughly analyzed, including the relevance of data indicators (e.g. df, F in Table 1). Instead of just writing out data sources and collation (such as Fisher's Exact Test in 4.2 and entire data in 4.5), they are better shown in the form of graphs or tables.
(3) Neither the quantitative analysis nor the conclusion are insightful. It is advised to highlight certain significant information in the conclusion.
(4) There is a spelling issue that has to be identified as British or American English. In the introduction, the word "recognize" is in British English, whilst the word "behaviour" in the abstract is in American English.
Author Response
Dear Reviewer,
Thank you very much for your helpful and constructive comments. Please find my point-by-point response to your comments below and the revised manuscript (Track Changes version) in the attachment. Please let me know whether more information is needed.
Response to Reviewer 3 Comments
Point 1: The logic of the paper has to be improved. The introduction and background parts should be combined, and the section on IEQ's influence should be somewhat condensed (introduction section). It is necessary to link information on behavioral [sic] toxicity, smoking, carbon dioxide, and other topics with cues and, ideally, to see a clearer contextual framework (e.g., the impact of acoustic environmental quality, air quality, lighting quality, and thermal comfort on mental health). The research state and current issues should be summed up at the end.
Response 1:
-The introduction and background parts were combined. (Page 2, Line 67)
- Some parts of IEQ’s influence were removed and the introduction section was condensed. (P1-2, L45-65)
- The impact of acoustic environmental quality, air quality, lighting quality, and thermal comfort on mental health has been addressed as follows:
Several behavioural toxins and hazardous materials (e.g. lead, mercury, and manganese) impact psychological wellbeing and are linked to the symptoms of anxiety and depression [44]. (P2, L94-96)
Research has found a dose-response relationship between traffic noise and psychological distress among children [55]. It has also revealed that airport noise adversely influenced children’s and adults’ psychological wellbeing [56, 57]. (P3, L113-116)
In this regard, Evans [64] showed that levels of illumination, especially the amount of daylight exposure, affect psychological wellness. (P3, L126-127)
A few studies have examined the links between mental health and thermal comfort. In a study on European housing, Bonnefoy et al. [79], found a relationship between depression/anxiety and residing in a home with insufficient protection against exposure to external factors, such as cold and draughts, in addition to noise and lack of light. (P3-4, L151-154)
- The current issues and research state were summed up at the end as follows:
Children spend most of their time indoors while most existing housing has been designed and built without considering their needs, especially for those with different cognitive abilities such as children diagnosed with ADHD who are extra sensitive to impacts from their everyday surroundings. Therefore, this study aimed to identify housing indoor environmental quality that may adversely impact children’s ADHD symptoms and behaviours. (P9, L343-348)
Point 2: There is no data analysis in place. Tables 1, 2, and 3's data haven't been thoroughly analyzed [sic], including the relevance of data indicators (e.g. df, F in Table 1). Instead of just writing out data sources and collation (such as Fisher's Exact Test in 4.2 and entire data in 4.5), they are better shown in the form of graphs or tables.
Response 2:
- More information has been added to the data analysis by presenting data in the form of graphs in Appendices A, B, C, and D, attached at the end of the manuscript. (P10-23, L436-514)
Point 3: Neither the quantitative analysis nor the conclusion are insightful. It is advised to highlight certain significant information in the conclusion.
Response 3:
- New information has been added to the quantitative analysis in the form of graphs presented in Appendices A, B, C, and D. (P10-23, L436-514)
- The Discussion and Conclusion sections have been improved, and significant information highlighted as follows:
Some of our results were congruent with prior studies such as the findings about the importance of air, lighting, and acoustic quality for children with ADHD. In support of our findings, previous research has linked the development of neurological disorders, learning disabilities, and cognitive deficits (such as ADHD and ASD) with exposure to environmental pollutants and toxic chemicals [45-47].
Consistent with our findings, studies have also shown that artificial light [67] and daylight [69] affect children’s behaviour and concentration ability. Although natural lighting is one of the important elements to achieve adequate lighting in the area for ADHD and ASD [83], as also shown in our findings, too much daylight or direct natural lighting (glare) can increase distraction and affect a child’s performance, especially at school [63, 77, 78].
In line with our findings, former research has also shown that low acoustic quality and noise adversely influenced children’s psychological wellbeing [55, 56] and caused inattention and misbehaviour in children with ADHD and ASD [65, 77, 83].
Some of the research outcomes are novel and considered as an initial step that makes a significant contribution to existing knowledge. In this regard, our research revealed that maintaining thermal comfort in the home environment is more significant for children with ADHD which has been poorly covered in the literature, if not neglected. Thus, our findings suggest that besides having suitable acoustic quality, air (such as air free from unpleasant smell or dust), and sufficient lighting (avoiding too much daylight or insufficient artificial lighting), it is important to use a heater and fan to provide a comfortable temperature within the home environment. Morning daylight and using a vacuum cleaner for cleaning the home were also found important. (P9, L364-386)
In conclusion, this research has provided insights into the importance of housing features and suggests that improving housing indoor environmental quality, mainly thermal comfort, air, and lighting quality, could positively correlate with alleviating ADHD symptoms and severity among children and adolescents. (P10, L398-407)
Point 4: There is a spelling issue that has to be identified as British or American English. In the introduction, the word "recognize" is in British English, whilst the word "behaviour" in the abstract is in American English.
Response 4:
- The spelling has been made consistent with the words “Recognise” and “Behaviour” now spelt in British English. (P1, L13-16 & P2, L56)

Round 2
Reviewer 1 Report
My previous comments have been well addressed in the revised manuscript. Therefore, I recommend the publication of this article on Sustainability.